



# Simulating soil organic C dynamics in managed grasslands under humid temperate climatic conditions

Asma Jebari[1], Jorge Álvaro-Fuentes[2], Guillermo Pardo[1], María Almagro[1], Agustin del Prado[1]

[1]Basque Centre for Climate Change (BC3), Edificio Sede no. 1, Planta 1, Parque Científico de UPV/EHU, Barrio Sarriena s/n, 48940 Leioa, Bizkaia, Spain

[2]Estación Experimental de Aula Dei (EEAD), Spanish National Research Council (CSIC), Av. Montañana, 1005, 50059 Zaragoza, Spain.

*Correspondence to*: Asma Jebari (asma.jebari@bc3research.org)

**Abstract.** Temperate grasslands are of paramount importance in terms of soil organic carbon (SOC) dynamics. Globally, research on SOC dynamics has largely focused on forests, croplands and natural grasslands, while intensively managed grasslands has received much less attention.

In this regard, we aimed to improve the prediction of SOC dynamics in managed grasslands under humid temperate regions. In order to do so, we modified and recalibrated the SOC model RothC, originally developed to model the turnover of SOC in arable topsoils, which requires limited amount of readily available input data. The modifications proposed for the RothC are: (1) water content up to saturation conditions in the soil water function of RothC to fit the humid temperate climatic conditions, (2) entry pools that account for particularity of exogenous organic matter (EOM) applied (e.g., ruminant excreta), (3) annual variation in the carbon inputs derived from plant residues considering both above- and below-ground plant residue and rhizodeposits components as well as their quality, and (4) the livestock treading effect (i.e., poaching damage) as a common problem in humid areas with higher annual precipitation. In the paper, we describe the basis of these modifications, carry out a simple sensitivity analysis and validate predictions against data from existing field experiments from four sites in Europe. Model performance showed that modified RothC reasonably captures well the different modifications. However, the model seems to be more sensitive to soil moisture and plant residues modifications than to the other modifications. The applied changes in RothC model could be appropriate to simulate both farm and regional SOC dynamics from managed grassland-based systems under humid temperate conditions.

## 1 Introduction

Temperate grasslands, which cover $1.25 \times 10^9$ ha globally, are important sinks of SOC, containing approximately 12% of the global SOC pool (Lal, 2004). Changes in grassland management (e.g., stocking rate, fertilisation) are frequent in temperate conditions affecting SOC dynamics (Soussana et al., 2004). Moreover, livestock rely on pastures, which constitute 86% of its total diet (Mottet et al., 2017). Therefore, improving the methods to estimate SOC stock changes in managed grasslands is key to obtain reliable estimates of SOC (Herrero et al., 2011) and determine the real contribution of livestock to the net global greenhouse gas (GHG) emissions.



Recent research in temperate grasslands has shown that grasslands can act either as C sink (Ma et al., 2015; Eze et al., 2018a) or source (Abdalla et al., 2018) depending on how animals, vegetation, soil, climate, and management practices interact with

each other (Rees et al., 2013;  Soussana et al., 2013; Graux et al., 2013). Assessing the climatic relevance of these SOC changes is however, challenging, as, SOC changes need to be evaluated within a sufficiently long time-frame (i.e. decades). Therefore, this type of studies requires long-term field trials (e.g. Skinner and Dell, 2014; Gourlez de la Motte et al., 2018), which are costly to maintain. Alternatively, as a way to study these dynamics, we can use simulation models to obtain complementary information to trials, for example, hypothesis forming or/and to predict long-term responses of grasslands to external factors

such as climate change and management (FAO, 2018). Models vary in complexity depending on their fundamental objectives (Taghizadeh-toosi and Olesen, 2016). For strategic studies, e.g. assessing potential of grasslands to sequester SOC in a region, simple soil models, e.g. RothC (Coleman and Jenkinson, 1996), ICBM (Andren and Katterer, 2016), C-Tool (Taghizadeh-Toosi et al., 2014) and Yasso07 (Tuomi et al., 2009) are most useful as they require a limited number of input data and easily available. Amongst these models, the RothC model is one of the models that has been most widely validated and effectively

used for different agricultural systems at different spatial scales (e.g. Poeplau and Don, 2013; Senapati et al., 2013; Smith et al., 2014). RothC was originally developed for arable soils under a range of soil and climatic conditions and hence, it has been widely used and parameterized for these systems.

Grasslands ecosystems under humid temperate conditions are subject to processes that may differ from arable systems in regards with SOC sequestration. In particular, below-ground plant residues in grasslands provide important C inputs for soil

C sequestration (Sainju et al., 2017): Grassland species allocate more C below-ground than cereals (Pausch and Kuzyakov, 2018) and below-ground C has longer residence time than above-ground C (Cougnon et al., 2017). Furthermore, rhizodeposition is an important source of C inputs (Kuzyakov and Schneckenberger, 2004), which is rarely quantified and still remains the most uncertain component of soil C fluxes in terrestrial ecosystems (Pausch and Kuzyakov, 2018).

Furthermore, grazing and wheeling by vehicles can cause damage of the soil structure by trampling and poaching, which

affects plant production,  and the potential amount of C inputs causing soil C loss (Eze et al., 2018b)); Ma et al. (2016)). Under humid temperate conditions, precipitations are high and winters are relatively mild with a relatively long growing season, susceptible to poaching (Tuohy et al., 2014). Poaching is a common soil damage problem of livestock treading which has not been extensively simulated in grazing ecosystems (Miao, 2016). Also, humid temperate climatic conditions imply that soils are wet-saturated during certain wet periods in which decomposition of organic matter is limited (Moyano et al., 2013).

Studies using RothC for grassland ecosystems implied specific initialization (Liu et al., 2011; Nemo et al., 2017) using information from long term grassland experiments (Cagnarini et al., 2019). Despite the number of possible interactions in grassland systems, e.g. between plant, soil and animals, RothC simplified the effects of different management affecting some of these processes on grasslands. For example, RothC indirectly simulate grazing activity by altering the amount of plant residues. i.e. total plant C inputs, but it does not distinguish between above- and below-ground C inputs (Nemo et al., 2017).

For animals C inputs, it uses default quality values for C inputs from grazing animals or manure applications, but it does not consider the treading effect on soil physical conditions (Smith et al., 2014). For extrema soil water conditions, Farina et al.



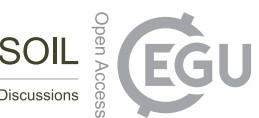

(2013) presented a modification of the model for dryland conditions in which reducing decomposition rate in soil improved model performance in these dry regions. However, for water-logged conditions RothC does not specifically considers that humid saturated conditions imply oxygen limitation and thus a decline in decomposition rate (Moyano et al., 2013).

Considering these potential factors that are not explicitly included in RothC, we studied which of the aforementioned factors (i) could be easily included in RothC, (ii) would affect SOC changes and (iii) could allow RothC improve predictions in SOC changes. For that, modifications performance was evaluated against available data from published experiments under humid temperate grassland ecosystems using a stepwise approach and through a sensitivity analysis.

## 2 Materials and methods

### 2.1 RothC model overview

The RothC -26.3 (Coleman and Jenkinson, 1996) model divides the SOC into five fractions, four of them are active and one is inert (i.e., inert organic matter, IOM). The active pools are: decomposable plant material (DPM), resistant plant material (RPM), microbial biomass (BIO) and humified organic matter (HUM). The decomposition of each pool (except IOM) is governed by first-order kinetics, characterized by its own turnover rate constant and modified by environmental factors related

to air temperature, soil moisture and vegetation cover, which are the main input parameters to run the model. Incoming plant C is split between DPM and RPM, depending on the DPM:RPM ratio of the particular incoming plant material or organic residue. Both of them decompose to produce BIO, HUM and evolved $CO_2$. The proportion that goes to $CO_2$ and to BIO + HUM is determined by the clay content of the soil which is another input to the model.

The model uses a monthly time step to calculate total SOC and its different pools changes on years to centuries time scale.

### 2.2 RothC tested modifications

The next four modifications were proposed and tested in this study: (i) soil water content function extended up to saturation; (ii) entry pools that account for the diversity of applied exogenous organic matter (EOM) from ruminant excreta; (iii) plant residue components and variability of its quality; and (iv) the poaching effect of grazing animals.

### 2.2.1 Soil water saturation in RothC

RothC assumes a minimum rate modifying factor for moisture when soil is at its minimum moisture capacity (i.e., at the extreme of water limitation) and not at the other extreme of oxygen limitation when soil is at its maximum capacity. In order to represent the reduction in the decomposition rate above field capacity (Moyano et al., 2013; Yan et al., 2016; Han et al., 2019), the rate modifying factor for moisture was assumed to follow a linear decline trend until a minimum rate of 0.2 (20%), at saturation conditions, as assumed by Smith et al. (2010) in the ECOSSE model.

The conversion from soil water content to soil moisture deficit (mm) used in RothC is given by the following equation (Eq. (1)) (Farina et al., 2013):



$$SMD_i = (WC_i - WC_{fc}) \times 10 \times depth \tag{1}$$

Where $SMD_i$ is the soil moisture deficit, $WC_{fc}$ is the soil water content at field capacity, $WC_i$ is the soil water content above field capacity.

Soil water contents at saturation and field capacity conditions are estimated considering soil properties related to soil texture (Raes et al., 2017).

### 2.2.2 Exogenous organic matter diversity (EOM)

Peltre et al. (2012) estimated EOM partition for the RothC model, based on an indicator of potential residual organic carbon in soils (IROC), which is derived from Van Soest fractions and the proportion of EOM mineralized during 3 days of incubation.

Similarly, Mondini et al. (2017) improved the prediction of SOC stocks in amended soils by fitting the RothC partitioning pools of different EOM to the respiratory curves. Such adjustment of the partition of EOM into RPM, DPM and HUM entry pools of RothC provided a successful fit and had been reproduced in other studies (e.g., Pardo et al. (2016)). However, the above-mentioned studies sum up all the different animal excreta into one category and did not distinguish excretions from different types of animal (e.g., ruminants, pigs…).

In order to capture the specific characteristics of ruminant excreta, we developed a methodology based on Pardo et al. (2017) as illustrated in Fig. A1. This study proposed a partition of the C inputs from excreta into RothC pools based on Van Soest fractions and the biodegradability of the material. A relationship between lignin content and anaerobic biodegradability is estimated as follows (Eq. (2)):

$$B = 0.905 \times exp(-0.055 \times lig(\%)) \tag{2}$$

Where B is biodegradability and Lig is lignin content as % of Volatile Solids (VS).

The Van Soest fractions are then partitioned into the pools of RothC based on its degradability, represented by the parameter B (i.e, lignin, holocellulose and solubles). A fraction of lignin is allocated into the HUM pool, representing the most resistant material. The rest of the lignin and the most resistant fraction of holocellulose and solubles are assigned to the RPM, while the most labile fraction of holocellulose and solubles are allocated to DPM.

This is expressed as VS %, following the equations (Eq. (3), Eq. (4) and Eq. (5)).

$$HUM = Lig \times (1 - B) \tag{3}$$

$$RPM = lig \times B + (Holocellulose + Solubles) \times (1 - B) \tag{4}$$

$$DPM = (Holocellulose + Solubles) \times B \tag{5}$$

The Van Soest fractions were derived from literature review for every animal excreta type of ruminants. However, it is worth to notice its variability (Fig. A2, Fig. A3) which depends on many factors, especially the diet (e.g., high concentrate diet implies lower lignin content in the ruminant´s excreta). In order to fit ruminant excreta quality to the RothC entry pools, we



opted for the average values for all fractions (Table 1). However, both extreme values (i.e., maximum and minimum) were assessed using a sensitivity analysis (See Sensitivity analysis section).

**Table 1** Ruminant excreta quality and its fitting to the RothC entry pools (based on scientific literature review).

|                  | RothC Pools |     |     |
| ---------------- | ----------- | --- | --- |
|                  | HUM         | RPM | DPM |
| Ruminant excreta | 0.1         | 0.6 | 0.3 |

### 2.2.3 Plant residue: components and quality

The RothC model does not distinguish between above- and below-ground plant residues. Model users generally use above-ground residues as a surrogate for total plant C inputs and they account less for root inputs (Nemo et al., 2017). Therefore, we

separated the plant residue C inputs into three components (i.e., above-ground residues, below-ground residues and rhizodeposits). The structure of C input derived from plant residues in RothC modified model is as illustrated in Fig. A4.

Parting from above-ground biomass, we used root to shoot (R:S) ratio to distinguish between above- and below-ground biomass and estimate below-ground biomass (when its value is not available). We assumed N fertilisation as the main driver for R:S ratio in grasslands as many studies have proved the strong dependence of the latter on N inputs (Poeplau, 2016 and

Sainju et al., 2017). We referred therefore to Poeplau (2016) equation (Eq. (6)) for RothC C input parameterisation under temperate grasslands in order to consider the fertilisation effect on the R:S ratio:

$$R:S = 4.7375\, e^{-0.0043\,.\,N\,input} \tag{6}$$

Where R:S is the Root: Shoot ratio and N input is nitrogen fertilisation expressed in kg N ha$^{-1}$ year$^{-1}$.

Unlike in annual croplands, in perennial grassland ecosystems, below-ground C biomass does not correspond to the below-

ground residue. Instead, below-ground residues correspond to 50% of the total below-ground C biomass (Poeplau, 2016) since the average annual root turnover of grasslands has been estimated to be 50 % in the temperate zone (Gill and Jackson, 2000). Regarding rhizodeposition estimation, we referred to an extensive literature review in which net rhizodeposition-to-root-ratio from grasslands was estimated to be 0.5 (Pausch and Kuzyakov, 2018).

We assumed a C concentration of 45 % of the plant biomass (Kätterer et al., 2012).

Plant residue quality (biochemical composition), as one of the main drivers of decomposition, is generally included in decomposition models (Zhang et al., 2008). Plant residue quality is represented in the RothC model by the DPM:RPM ratio (i.e., ratio of rapidly and slowly decomposing pools), which was obtained by optimization to obtain the best fit according to different land use types (Coleman, personal communication). For instance, for most agricultural crops and improved grasslands, RothC uses a DPM:RPM ratio of 1.44 (i.e. 59% of the plant material as DPM and 41% as RPM). For unimproved

grasslands and scrubs (including Savannas) a ratio of 0.67 is used (Coleman and Jenkinson 1999). The use of a unique ratio





may affect the reliability of model predictions since plant residue quality is variable and depends on several factors (e.g., maturity stage, climate variables and nitrogen fertilisation).

In order to fit the DPM:RPM ratio to the specific conditions of temperate grasslands, including its variability over the year, we used the stepwise chemical digestion (SCD) method (Goering and Van Soest., 1970). We adopted the same approach used by

Gunnarsson et al. (2008) and Borgen et al. (2011)) to assign the RothC pools to the SCD fractions. The DPM pool was considered as the C in the Neutral Detergent Soluble (NDS), and the RPM pool as the C in the Neutral Detergent Fiber (NDF) (i.e., holocellulose and lignin fractions). In the absence of NDF and NDS measured data, there are existing empirical existing equations that can help to have an estimation of these parameters. For our study we used an existing equation from Salcedo (2015), which was obtained for grasslands located in Northern Spain under humid temperate climatic conditions. This equation

empirically relates NDF% as a function of monthly air temperature, monthly water reserves and crude protein (Eq. (7)):

$$NDF\ (\%\ of\ dry\ matter)\ =\ 83.5\ -\ (1.27\ CP)\ -\ (0.35\ Tamean)\ -\ (0.005\ water\ reserves) \tag{7}$$

Where CP is crude protein and is expressed as a percentage of dry matter (CP is variable and depends on the stage of plant growth. It was obtained according to grassland plant species and their growth stage); $T^a_{mean}$ is the monthly air temperature in °C; Water reserves refer to the difference between monthly precipitation and potential evapotranspiration.

Regarding below-ground plant material quality, the quantity of lignin itself is the main potential driver of differential degradation between both above- and below-ground plant components (Rasse et al., 2005). Therefore, we added up the difference of lignin percentage of ~ 8% (between above- and below-ground parts) to get the below-ground RPM pool, referring to De Neergaard et al. (2002).

The DPM pool is then derived by subtraction according to the equation (Eq. (8)):

$$DPM\ (\%)\ =\ 100\ -\ RPM\ (\%) \tag{8}$$

Finally, we assumed that the C inputs derived from rhizodeposition are transferred to DPM of the RothC because of the expected rapid decomposition of this labile substance by rhizosphere microorganisms (Klosterhalfen et al., 2017).

### 2.2.4 Animal treading effect: Poaching

We developed a poaching sub-model according to data obtained from studies performed under humid temperate conditions

(Tuñon et al., 2014; Phelan et al., 2013; Tuohy et al., 2014). The aim of the sub-model is to detect the grade of soil damage and its impact on plant C inputs depending on soil moisture and grazing conditions (i.e., stocking rate) (Fig.1). Soil moisture is estimated in RothC using the Soil Moisture Deficit (SMD) value that considers rainwater inputs and soil texture properties (i.e., clay content). We used SMD as a proxy for soil moisture to predict when soil water conditions are likely to lead to hoof damage according to Piwowarczyk et al. (2011) and Herbin et al. (2011). For simplification reasons, we assumed water

saturation conditions from SMD = -10 mm onwards, as a condition of poaching occurrence as in Scholefield and Hall (1985) and Tuohy et al. (2014). Livestock density has an effect on the level and extent of treading damage (Nie et al. 2001; Tuohy et al., 2014) illustrated by hoof print depth and soil surface deformation (Tuñon et al., 2014). Depending on the soil surface deformation of a treading event, the pasture production is reduced (Phelan et al., 2013) and thus its plant C input (Eze et al.,





2018b) (Fig. 1, Fig. A4). The main equations related to the conceptual sub-model of poaching effect are described in Appendix
A.

As the poaching effect in temperate grazing systems seems to cause only short-term pasture reduction but there is a relatively
fast recovery to these damages (Black, 2014 and Tuñon et al., 2014), we considered that plant C input reduction due to poaching
effects would only occur only in the months with soil poaching.

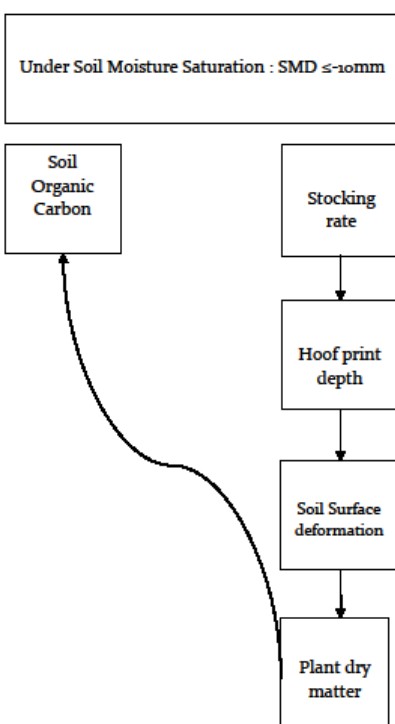

**Figure 1** Conceptual diagram of how the new RothC simulates the poaching effect impacts on SOC dynamics.

**2.3 Model validation**

**2.3.1 Study sites**

In order to validate the proposed modifications, we identified and used data from four studies located on different European
grasslands under humid temperate conditions and characterized by grass and clover mixture. The grassland sites (Laqueuille
intensive grazing grassland, Oensingen intensive cutting grassland, Easter Bush intensive grazing grassland and Solohead
dairy research farm) (Table B1) were mainly selected from the FLUXNET program (http://www.fluxnet.ornl.gov/; (Baldocchi,





2008). Information on geographic and climatic characteristics, soil properties and management of the different sites are listed in the Tables B1 and B2.  Detailed information on input data for the model and main assumptions are described in the Appendix B (Table B3).

**2.3.2 Model running and initialisation**

For RothC initialisation we used the pedotransfer functions established by Weihermüller et al. (2013) to estimate all active C pools from initial provided measured SOC stocks. The initial IOM pool was set to match the equation proposed by Falloon et al. (1998):

$$IOM = 0.049 \, TOC^{1.139} \tag{9}$$

We modelled SOC dynamics from the different study sites using a stepwise approach. First we used the default RothC version (RothC_0) and, subsequently we progressively added the different modifications tested (Table 2): (i) saturation conditions for soil water function (RothC_1 modification); (ii) ruminants excreta (RothC_2 modification); (iii) plant residue components and its characteristics (RothC_3 modification); and (iv) soil poaching (RothC_4 modification).

Soil organic carbon stocks were simulated at 20 cm depth for Laqueuille and Oensingen and at 30 cm depth at Easter Bush
and Solohead dairy farm.

**Table 2** RothC versions tested in the study with the modification included in each version.

| RothC version | Modifications |
| --- | --- |
| RothC_0 | RothC default version |
| RothC_1 | RothC_0 + saturation conditions for soil water function |
| RothC_2 | RothC_1 + ruminant excreta characteristics |
| RothC_3 | RothC_2 + plant residue characteristics and its variability |
| RothC_4 | RothC with all modifications: RothC_3 + inclusion of poaching effect |

**2.4 Model performance analysis**

The main objective of this study was to assess the ability of the proposed model modifications to predict the measured SOC
stocks from intensive grassland sites. For this, we used different performance indices and threshold criteria based on Smith and Smith (2007) (Table C1). The ability of each modification to improve SOC dynamics simulation was evaluated using the




relative root mean square error (RMSE), mean difference of simulations and observations (BIAS) and the model efficiency (EF) (Table C1).

## 2.5 Sensitivity analysis

Several studies have indicated that the RothC model is most sensitive to C inputs (Gottschalk et al., 2012; Stamati et al., 2013; Riggers et al., 2019). In our study, analyses were performed to test the sensitivity effect on SOC changes of the different modifications (other than C inputs) implemented in the model, using RothC_4. Model sensitivity was expressed as an index, which considered different input values related to the modifications (i.e., plant residues quality, ruminant excreta quality and soil moisture up to saturation) from minimum to maximum (Table 3) and then the output values were analysed according to

the following index (Eq. (10)) (Smith and Smith, 2007).

$$Sensitivity\ index = \frac{max(Pi) - min(Pi)}{max(Pi)} \tag{10}$$

Where max (Pi) is the maximum output value and min (Pi) is the minimum output value.

We used NDF as a proxy for RPM in relation with plant residues quality (Table 3), assuming that NDF varies from 30 to 70% as minimum and maximum values based on 15 papers (Table D1). We used the lignin fractions (% VS) as a proxy for EOM

in relation with ruminant excreta quality assuming minimum and maximum values from literature values shown in Table 3. Similarly, for soil moisture variation, we tested minimum (0.2) and maximum values (1) of the rate modifying factor for moisture (Table 3).

**Table 3** Model modified components used for the sensitivity analysis and their interval values.

| Modified component | Proxy | Interval for possible values |
|---|---|---|
| Plant residues quality (e.g., perennial grass) | NDF | 30 – 70 % |
| Quality of ruminant excreta (e.g., cattle slurry) | Lignin | 9 – 28% |
| Soil moisture up to saturation | Rate modifying factor for moisture | 0.2 – 1 |




## 3 Results and discussion

### 3.1 Measured versus simulated SOC stocks

All four sites showed a similar pattern of SOC changes with the RothC default version (i.e., RothC_0) as well as with the four modified versions (Fig. 2). In all four sites, the lowest simulated SOC stocks were observed in the default model version (RothC_0). RothC_0, for Laqueuille, Oensingen and Solohead sites, simulated that SOC was reduced during the time of the experiment (Fig. 2a, Fig. 2b and Fig. 2d), which was the opposite trend that measurements showed. For example, in the Laqueuille intensive grassland, SOC stocks predicted by the RothC_0 version decreased from 114 to 102 Mg C ha$^{-1}$ whereas measured values increased from 114 to 121 Mg C ha$^{-1}$ (Fig. 2a).

By introducing the soil moisture modification in RothC (RothC_1), the model simulated an increase in SOC stocks (Fig. 2). For example, SOC stocks at the end of the simulation period in 2011 reached 86.5 Mg C ha$^{-1}$ (RothC_1) compared to 83.7 Mg C ha$^{-1}$ (RothC_0) in the Easter Bush intensive grazing grassland (Fig. 2c). Soil moisture modification at saturation reduces decomposition rates values for very wet soils conditions. In fact, the 4 sites used in our study have soil water saturation during many months of the year.

By implementing changes to account for ruminant excreta quality (RothC_2) on the study sites, the model also resulted in an increase in SOC in time, which, trend-wise, differs from the RothC_0 model, but coincides with measured data. Moreover, this SOC increase was lower than that simulated by RothC_1 (Fig. 2). Changes in the modification of plant residues (RothC_3) resulted in greater SOC increased values in time when compared with the previous modification (RothC_2) (Fig. 2). The lower effect of the simulation of animal excreta characteristics in RothC_2 could be explained by the higher quantity of plant residues while adding the rhizodeposition component together with above- and below-ground components in RothC_3.

Including the poaching effect (RothC_4), resulted in slightly reduced SOC stocks compared with RothC_3, specially for the Easter Bush and Solohead sites (Fig. 2a, Fig. 2c and Fig. 2d). This reduction in SOC stocks in RothC_4 compared with the RothC_3 version could be explained by the reduction in plant C inputs due to poaching that typically occurs at saturation conditions (Menneer et al., 2005; Eze et al., 2018b).

In general, the highest predicted SOC stocks values and the closest to the measured values resulted after RothC_3 and RothC_4 simulations (Fig. 2).





(a)

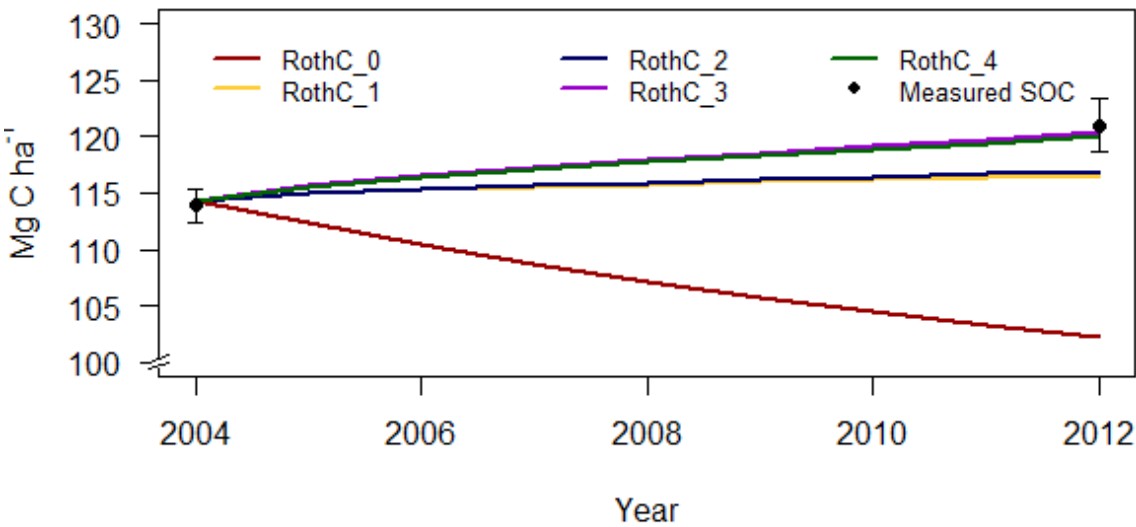





(b)

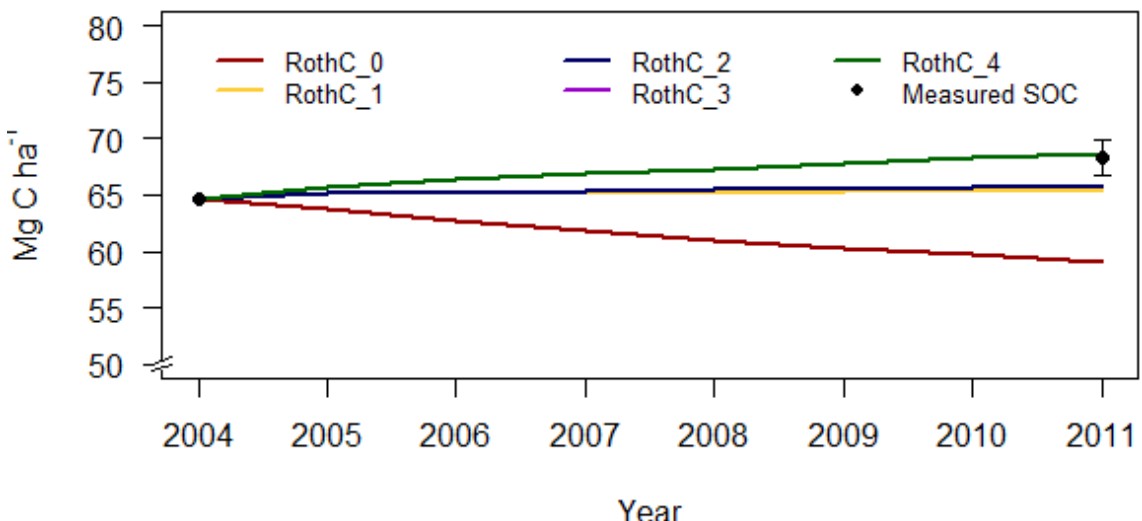







(c)

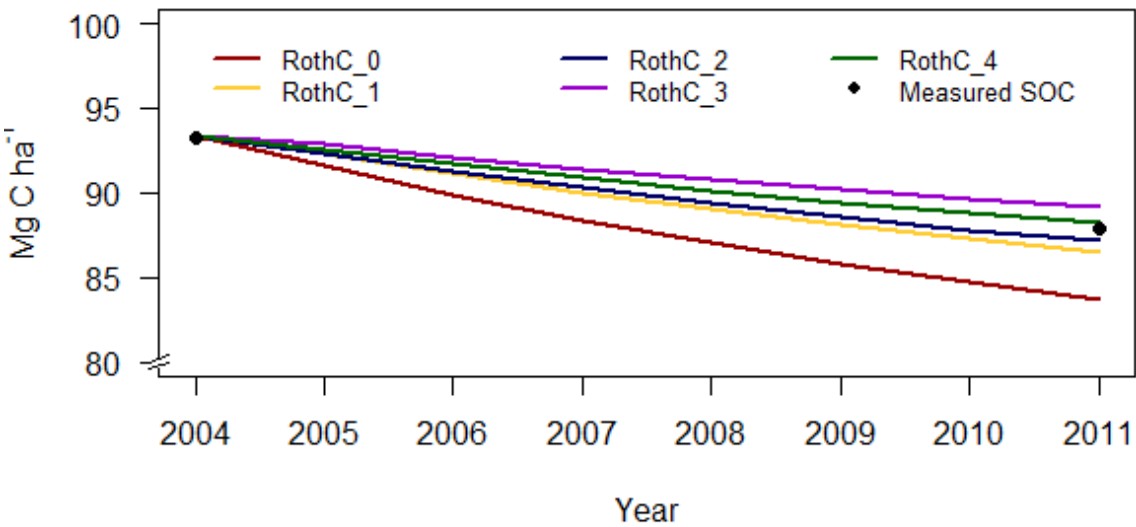





(d)

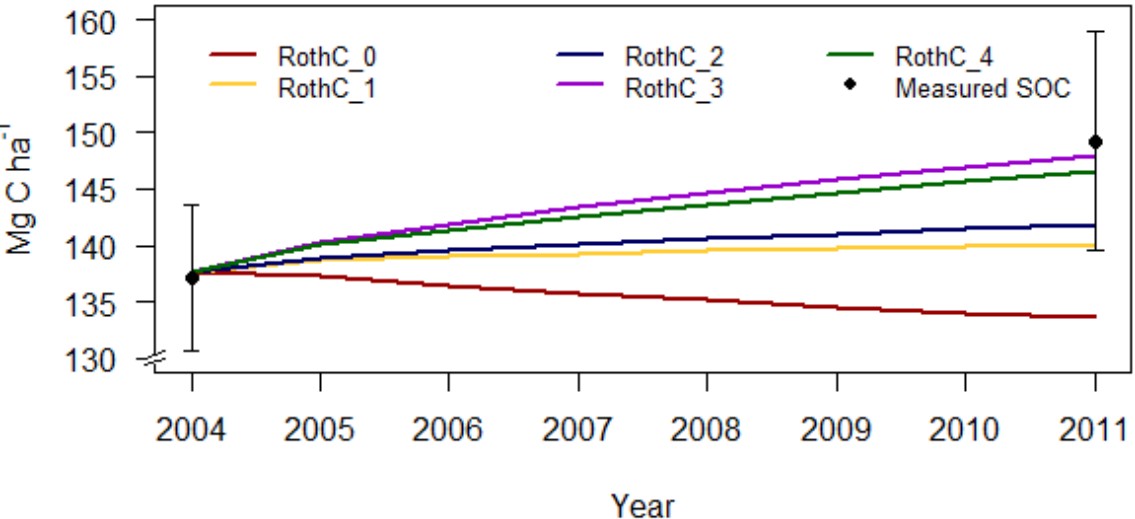

**Figure 2** Measured and simulated SOC stocks (Mg C ha$^{-1}$) using the default RothC model (RothC_0) and the modified RothC
versions (RothC_1; RothC_2; RothC_3; and RothC_4) for the different validation sites: (a) Laqueuille intensive grazing
grassland; (b) Oensingen intensive cutting grassland; (c) Easter Bush intensive grazing grassland; and (d) Solohead dairy
research farm.

**3.2 Model performance**

In general, the RothC default version (RothC_0) showed a good performance with an EF value of 82% (Table 4). However,
the different modifications presented enhanced the predicting performance of RothC for these specific sites. In particular,
simulated SOC stocks using the RothC_3 and RothC_4 versions almost matched measured values (Fig. 3) achieving model
efficiencies of 99% (Table 4). Therefore, these two modifications accurately predicted SOC changes.



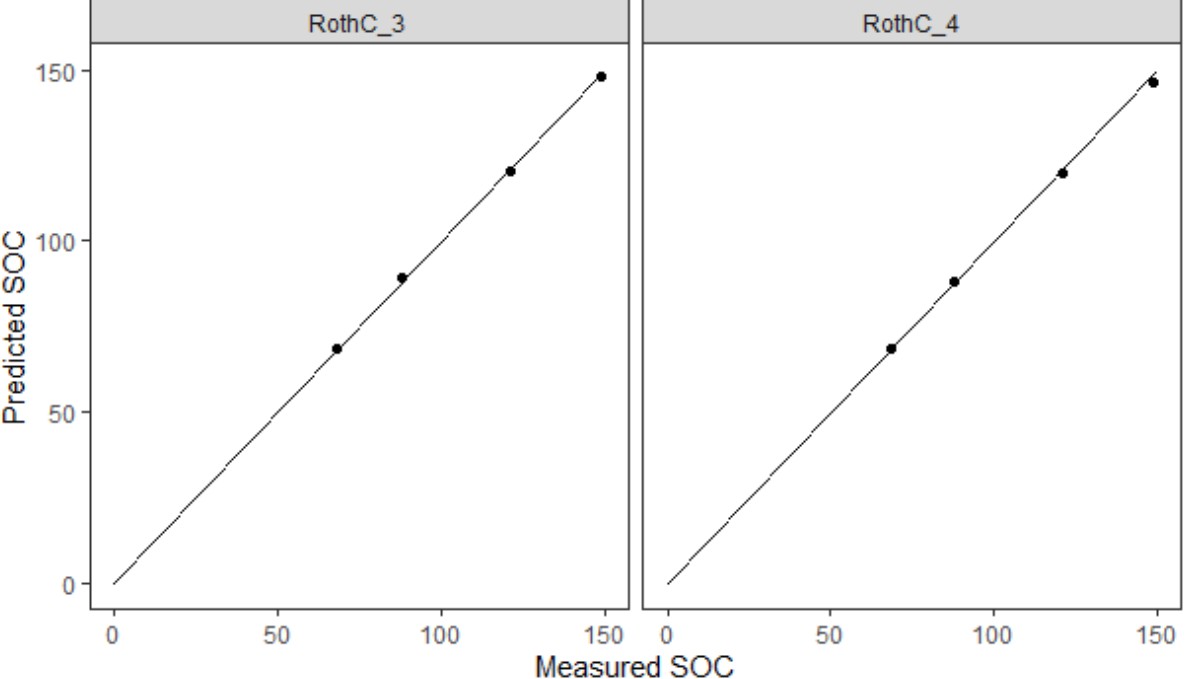

**Figure 3** Measured versus predicted values of SOC stocks at the end of the simulation period using RothC_3 and RothC_4 model versions for the different study sites.

The negative bias (reaching -18.8 in Laqueuille site) and the higher RMSE values obtained in RothC_0 compared with the different RothC modified versions indicated that the default RothC version underestimated SOC stocks, especially in the Laqueuille and Solohead sites, which presented the highest SOC content (Table 4). The modification of the soil moisture function in RothC_1 reduced the bias and the RMSE (Table 4) and improved the general trend of SOC stocks compared with the default version RothC_0 in all simulated sites (Fig. 2). RothC_0 assumes high decomposition rates with high soil moisture, but it does not consider the cessation of the decomposition process which occurs in high wet soils close to saturation conditions (Das et al., 2019), frequent in humid temperate grasslands. The inclusion of the ruminant excreta quality in the model only slightly improved the SOC predictions in RothC_2 compared to RothC_1 (Table 4). In this context, Heitkamp et al. (2012) and Mondini *et al*. (2017) emphasised the importance of modifying the quality of residues to improve the model performance, concluding that the adjustment of DPM:RPM ratio led to better model performance than the use of default DPM: RPM values provided by the model.

Comparing RothC_2 and RothC_3 versions, it could be deduced that integrating quantity and quality distinction of plant residue in RothC_3, as a primary source of SOC (Castellano et al., 2015), improved SOC predictions. Adding the modification of plant residues in terms of quantity and quality contributed to improve SOC simulation compared to the modification of specifying animal excreta quality. The improvement showed by plant residues modification could be explained by the higher sensitivity of the model to C inputs quantity compared to C inputs quality and the importance of including rhizodeposition



together with above- and below-ground components in plant C input quantification as in RothC_3. Indeed, as a fundamental
source of C inputs, rhizodepostion is rarely quantified and remains the most uncertain component of the soil C fluxes (Pausch
and Kuzyakov, 2018).

The poaching effect is assumed to reduce plant productivity and the potential amount of C inputs to the soil (Eze et al., 2018b)
and thus causing SOC loss (Ma et al., 2016). Consequently, the poaching sub-model included in the RothC_4 version predicted
reductions in SOC stocks compared to the RothC_3 version (Fig. 2a, Fig. 2c and Fig. 2d). The reduction in SOC stocks is
explained by the lower C inputs during the months when grazing occurs under saturation conditions. Only in the case of the
Easter Bush site, the poaching sub-model contributed to improve SOC predictions in the RothC_4 version (Table 4, Fig. 2c).
A possible explanation to this improvement in the SOC predictions is that the soil in Easter Bush site is poorly drained and
grazing by ruminants occurs all year round and thereby highly susceptible to poaching. In the same context, Vuichard et al.
(2007) enhanced the original PASIM grassland constructing a simple and empirical model of the detrimental impact on
vegetation of trampling by grazing animals by removing at each time step a fixed proportion of the above- ground biomass.
However, it is important to point out the complexity of the poaching effect, as it induces more impacts other than the
detrimental vegetation impact which are beyond the scope of our study. Moreover, more robust experiments are needed in
order to define the severity of the poaching effect according to soil moisture saturation, livestock density and soil type.

In general, RothC_3 and RothC_4 versions showed the lowest bias and RMSE values of all the different versions tested and
the best agreement with field measured data (Table 4, Fig. 3). Effects of the different modifications tested were more noticeable
in the Solohead and Laqueuille grassland intensive sites, which showed higher SOC and saturated conditions during the year.
In RothC_4, we considered different variables (i.e., soil texture, precipitation, grass type, grazing intensity, study duration and
sampling depth), which significantly influenced the variation in grazing effects on SOC (Mcsherry and Ritchie, 2013).

The model modifications with the greatest positive impact were soil moisture and plant residues (Table 4, Fig. 2). In contrast,
the effect of animal excreta quality and poaching on SOC simulation by RothC was low.

**Table 4** Root mean square error (RMSE) and mean difference of simulations and observations (BIAS) for each model version
and grassland intensive site and model efficiency (EF) and RMSE across sites.

| Site | Performance test | RothC_0 | RothC_1 | RothC_2 | RothC_3 | RothC_4 |
|---|---|---|---|---|---|---|
| Laqueuille | BIAS | -18.83 | -4.49 | -4.04 | -0.64 | -1.02 |
| Oensingen | BIAS | -9.22 | -2.95 | -2.61 | 0.32 | 0.32 |
| Easter Bush | BIAS | -4.13 | -1.33 | -0.74 | 1.30 | 0.35 |





| Solohead | BIAS | -15.63 | -9.13 | -7.32 | -1.23 | -2.7 |
|----------|------|--------|-------|-------|-------|------|
| All sites | RMSE | 12.42 | 5.01 | 4.13 | 0.90 | 1.37 |
| | EF | 0.82 | 0.97 | 0.98 | 0.99 | 0.99 |

## 3.3 Sources of uncertainty

Although RothC_3 and RothC_4 simulations performed well in simulating SOC changes for the selected sites, there were
limitations related to the uncertainty of, both, model inputs and modifications, and to the limitation of the data used for
validation.

Regarding model inputs, uncertainty was mainly related to the lack of detailed measured data of C inputs derived from plant
and/or animal origin. In this study, we used the average of available measured values (details can be found in the section "Input
data for the model and main assumptions" in Appendix B). However, measured C inputs is not always available, so its value
could be supplied via linkage with another model, considering the grazing effect (case of plant residues). It is important to
point out that previous studies running RothC in grassland ecosystems overestimated C inputs (Nemo et al., 2017) and there
is a lack of detailed information on how plant residues were estimated and/or assumptions regarding their conversion to C
inputs (Nemo et al., 2017). In particular, the estimation of below-ground C inputs is another major source of uncertainty for
SOC modelling (Keel et al., 2017). Indeed, belowground C inputs depend on multiple factors, including site-specific
agronomic practices and the response of plant genotypes to them, and direct measurements of belowground C inputs is a
challenging issue (Cagnarini et al., 2019). Moreover, the use of pedotransfer equations for initialising SOC pools, as an
alternative for soil physical fractionation, may represent another source of uncertainty (Van Looy et al., 2017). Indeed, although
the reliability of pedotranfer equations, they could reveal some errors which are in the range of measurement error for SOC
(Weihermüller et al., 2013).
Regarding model modifications, a linear decline in the rate modifying factor for soil moisture was assumed under saturation
conditions, like in the ECOSSE model, as there was not sufficient evidence to suggest a more refined relationship as indicated
by Smith *et al*. (2010). However, the effect of soil moisture on SOC dynamics is complex and non-linear (Batlle-Aguilar et
al., 2010), interacting with temperature effect (Lee et al., 2018). Improvements could be achieved by using a more refined
function based on robust field experiments. Furthermore, our estimations of animal excreta quality are not conclusive and
further refinements based on experiments could be made as, for example, to account for animal intake quality to predict its
excreta quality. Furthermore, with reference to plant residues, the use of empirical equation to estimate NDF for plant residue
quality is subject to uncertainty as it relies on different dynamic variables (i.e., Temperature, water reserves and CP). Regarding
the poaching effect sub-model, based on the literature review we made, the number of long-term experiments under humid

 

temperate region is limited. Moreover, due to the complexity of the soil damage (i.e., poaching) in which many factors could
be involved (i.e., soil, animal, plant) (Tuñon, 2013), it is difficult to generalise our findings. The lack of usable, mechanistic
simulation models of soil deformation under hooves and wheels is partly due to the lack of appropriate conceptual
understanding and theory of the complex soil mechanical processes involved (Scholefield et al, 1986). The other reason is the
shortage of good and relevant experimental data with which to form hypotheses and test any models.

In our study, simulations of the different modifications were compared to measured data of different study sites. However,
field measurements also have deviations, which are source of uncertainty as they are used as the scale to evaluate model
performance (Chen et al., 2017).

## 3.4 Sensitivity analysis

A sensitivity analysis of RothC_4 was performed to assess the robustness of the modifications (plant residues quality, ruminant
excreta quality and soil moisture up to saturation) made in the different model versions presented. In general, RothC_4 seems
to be more sensitive to C input quantity than to quality and to soil moisture function, particularly at saturation conditions.
The sensitivity analysis performed for resistant plant residues pool with the RothC_4 version showed a sensitivity index of
0.8% for the Easter Bush site and 2.6% for Oensingen and Solohead research farm (Table D2). In this context, Fallon (2001)
showed that the default RothC version is rather insensitive to variations in C inputs quality as varying DPM:RPM ratio for
plant residues from 0.1 to 2.0 (20-fold variation) resulted in a SOC decline from 29.0 to 24.3 t ha$^{-1}$ (about 16% variation).
However, according to other studies (e.g. Shirato and Yokozawa (2006); Heitkamp et al. (2012) and Borgen et al. (2011)),
specifying plant C input quality depending on residues partitioning instead of using the default RothC ratio for DPM and RPM
should enable more reliable modelling of SOM dynamics.
In relation to the sensitivity of the RothC_4 version to the animal excreta quality, the values of sensitivity index obtained for
the different experiments were in general low (between 1.1% and 3%) (Table D3). So, the use of average value for the different
animal excreta fractions does not really impact the results, as we implemented in EOM modification. The greatest value for
the Solohead research farm could respond to the higher C inputs derived from animal excreta that received Solohead research
farm as compared to the other sites. In order to focus on RothC_4 sensitivity to animal excreta quality with relation to its
quantity, we assumed an annual C input derived from animal excreta of about 2.5 t C ha$^{-1}$ distributed between March and June
for the remaining sites that receive smaller amount of organic fertiliser. As animal excreta quality in the RothC model is
connected to its quantity, the sensitivity index of animal excreta quality increased as its quantity increased (Table D3). In this
context, according to Mondini et al. (2018), RothC displayed a moderate sensitivity to variations in animal excreta quality,
more specifically the ratio between decomposable and resistant pools.
Sensitivity index regarding soil moisture modification was higher compared with the other modifications reaching, for example
12.8% in the Laqueuille site (Table D4). Therefore, the modified model is sensitive to the rate modifying factor for soil
moisture up to saturation under humid temperate climate conditions. In this context, Bauer et al. (2008) concluded that reliable



prediction of carbon turnover requires that the soil moisture together with the soil temperature reduction functions of the model need to be valid for the environmental conditions.

## 4 Conclusions

To our knowledge, this study is the first attempt to integrate grassland management effect under humid temperate conditions
into the RothC model to simulate SOC changes. The proposed modifications to the model considered the incorporation of distinction for plant residues components (i.e., above- and below-ground residues and rhizodeposition) in terms of quantity and quality and distinction for ruminant excreta quality, the extension of soil moisture up to saturation conditions and, finally, the introduction of the livestock damaging effect (i.e., poaching) on plant residues under water saturation conditions. The moisture response modification and the introduction of plant residues components were a major improvement in the model,
but plant residues and ruminant excreta quality had a small impact on model predictions. The poaching effect is complex and multi-factorial and our modification did not result in large differences to the original RothC. These modifications do not impair the performance of the model under temperate conditions and so represent a broadening in the capability of the RothC model to be more adapted to grassland-based livestock systems under humid temperate conditions. It must be kept in mind that although there was good agreement between results from modified model and measured data from different studies, validating
against more sites would greatly improve our confidence.

The modifications presented here to the RothC model would improve assessments of SOC changes in grasslands under temperate humid climatic conditions not only at a plot level but also at regional level. It could be a useful tool for stakeholders and policy makers in order to improve the quantification of SOC sequestration and develop effective strategies to reduce the impact of grassland-based livestock systems on global warming.








## Appendices

### A. Model modification

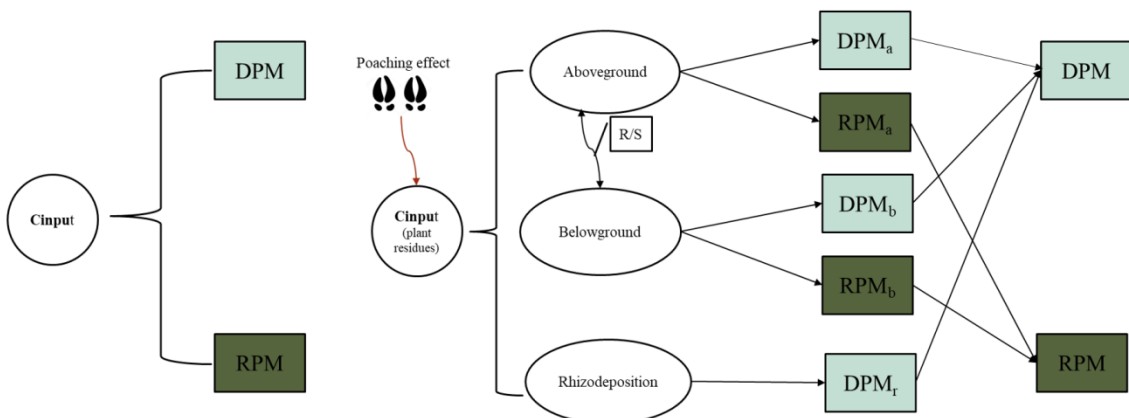


(DPM: decomposable plant material; RPM: resistant plant material; DPM$_a$, decomposable above-ground plant material; RPM$_a$, resistant above-ground plant material; DPM$_b$, decomposable below-ground plant material; RPM$_b$, resistant below-ground plant material; DPM$_r$, decomposable rhizodeposits)

**Figure A1** Structure of C input derived from plant residues in RothC modified model

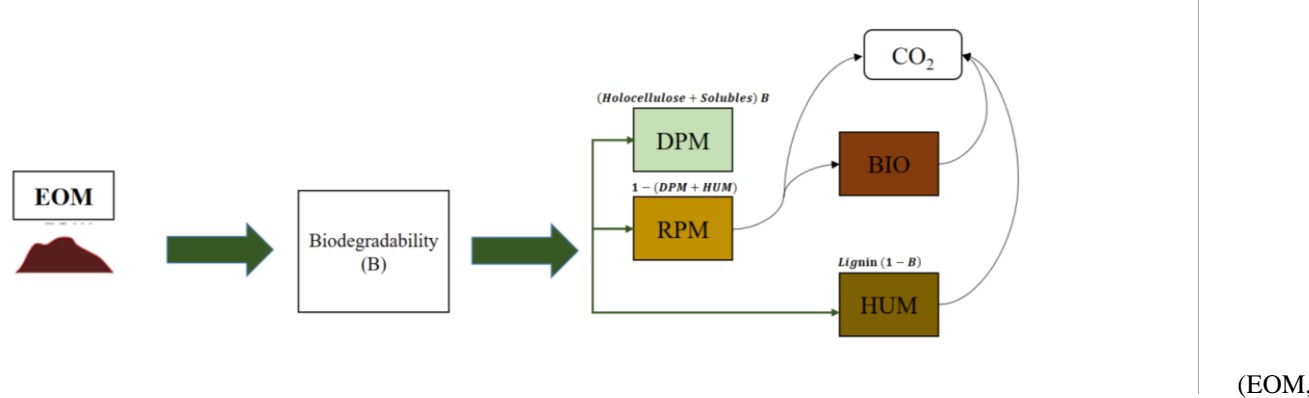

460                                                                                                              (EOM,

exogenous organic matter; DPM, decomposable EOM; RPM, resistant EOM; HUM, humified EOM)

**Figure A2** Structure of C input derived from EOM in RothC modified model




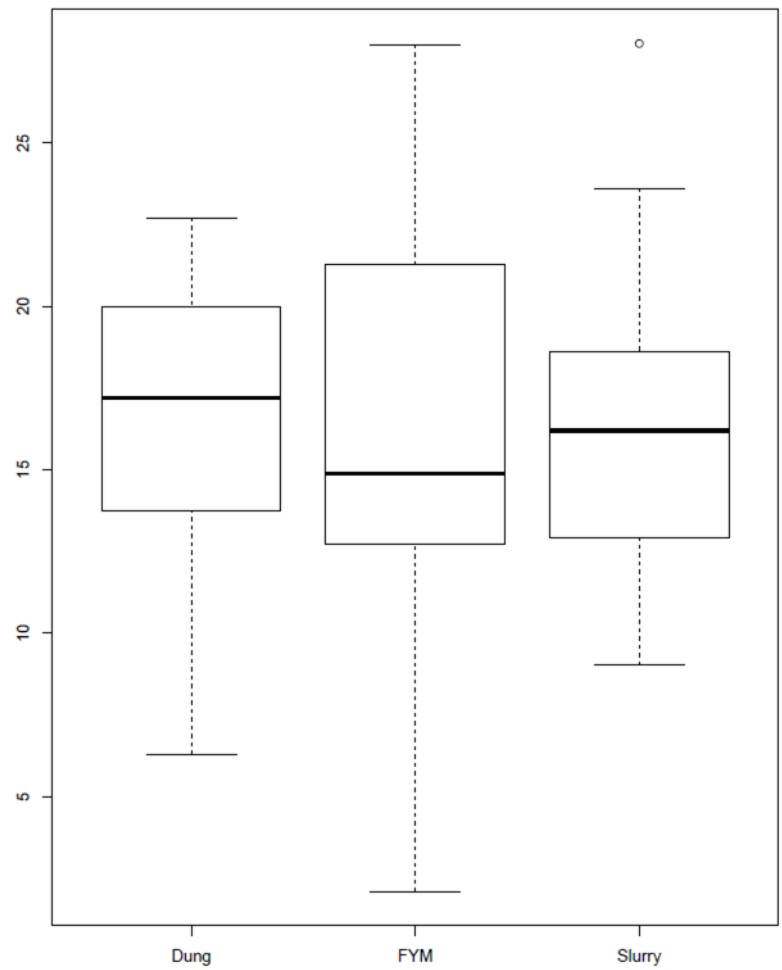

*FYM, Farmyard manure

**Figure A3** Boxplot displaying the lignin Van Soest fraction variability for the different ruminant residues' types (Dung, Farmyard manure, slurry), based on literature review findings





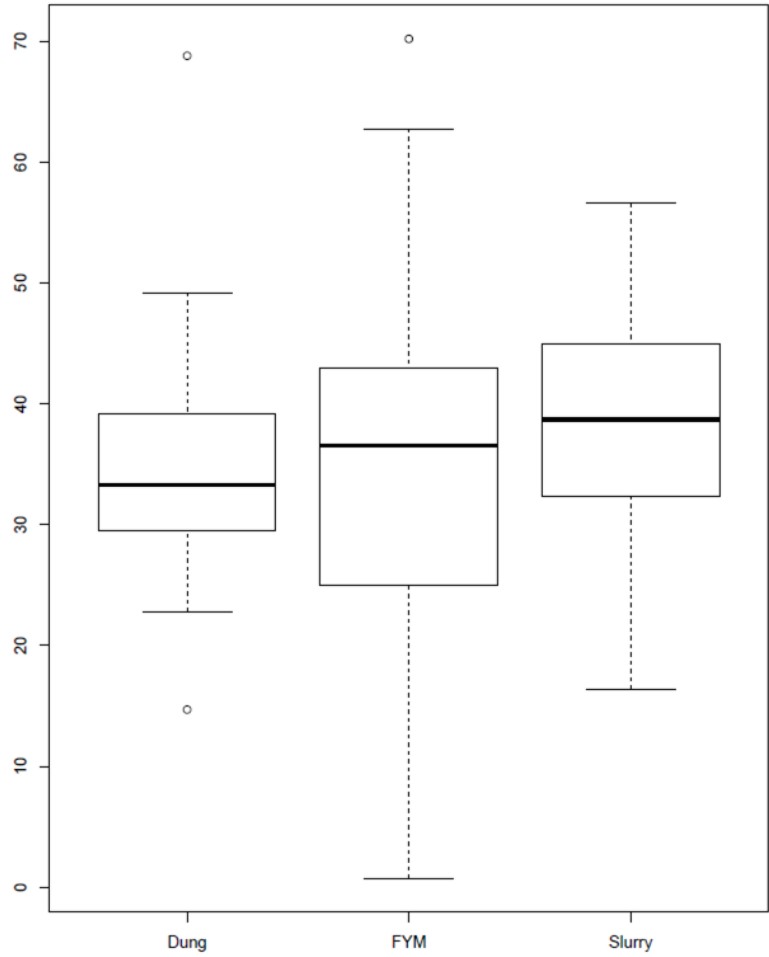


*FYM, Farmyard manure

**Figure A4** Boxplot displaying soluble Van Soest fraction variability for the different ruminant residues types (Dung, Farmyard manure, slurry), based on literature review findings

### Equations of cattle poaching effect sub-model

**Hoof print depth** (HPD) is function of stocking rate (SR) depending on the soil texture.

HPD is expressed in mm and stocking rate (SR) is expressed in number of cows/ha (Average live weight =550kg). Equations are deduced from experiments in Tuñon et al. (2014).

For example, for poorly drained soils: HPD = 33.072 (SR)$^{0.6927}$   $R^2$ = 0.77                    (A1)

**Soil surface deformation** (SSD) is significantly correlated with HPD (SSD=2·46 HPD+ 12·29 $R^2$=0·75)(A2)      (Tuohy et
al., 2014). It is expressed in m/m.



**The proportional reduction in herbage dry mass** (PRHM) following each treading event. This proportion is without unit and was applied for plant C input according to Phelan et al. (2013) equations.

PRHM = -2.58 SSD+8.55 $R^2$=0.49 (Early-spring turnout with an annual fertilizer input of 100 kg N ha$^{-1}$).    (A3)

PHRM = -1.78 SSD+2.56 $R^2$=0.51(Early-spring turnout with no Fertilizer-Ninput).    (A4)

PHRM = -2.92 SSD+17.56 $R^2$=0.62 (Late-spring turnout with no Fertilizer-Ninput).    (A5)

Where SSD is expressed in cm/m.

### B. Study sites description and input data

*1. Study sites description*

The Laqueuille intensive site is a semi-natural grazing grassland (2.81 ha). The soil is classified as Andosol (20% clay, 53%
silt and 27% sand) (FAO classification). The site was continuously grazed by heifers from May to October without additional feed supply (more details on Klumpp et al. (2011) and Ben Touhami et al. (2013)). The Oensingen intensive site is cutting grassland. The soil is classified as Stagnic Cambisol (Eutric) (FAO, ISRIC and ISSS, 1998). The field has been sown with grass- clover mixtures since 2001 and is mown 4 times per year and fertilised at the beginning of each growing cycle (Ammann et al. 2009). The Easter Bush experimental site is under permanent grassland grazing management. The soil is classified as
Eutric Cambisol (FAO classification) and is imperfectly drained. Grazing in this site occurs all year round by heifers in calf, ewes and lamb, which always have access to the entire field (more details on Skiba et al. (2013) and Jones et al. (2016). The Solohead site is a dairy research farm with poorly drained soils. From 2004 to 2011, a typical grassland management involved rotational grazing (Necpálová et al., 2013).

**Table B1** Location and climate of the grassland sites (available through the European Fluxes Database Cluster:
http://www.europe-fluxdata.eu (except Solohead farm))

| Site name | Country | Altitude (m) | Latitude | Longitude | Mean air temperature (°C) | Mean annual precipitation (mm) | Simulation period | References |
|---|---|---|---|---|---|---|---|---|
| Laqueuille | France | 1040 | 45° 38´N | 02° 44´E | 7 | 1012 | 2004-2012 | (Klumpp et al., 2011) |
| Oensingen | Switzerland | 450 | 47° 17´N | 07° 44´E | 9 | 1263 | 2004-2011 | (Ammann et al., 2009) |
| Easter Bush | United Kingdom | 190 | 55° 52´N | 03° 02´W | 9 | 1031 | 2004-2011 | (Skiba et al., 2013) |





| | | | | | | | | (Jones et al., 2016) |
|---|---|---|---|---|---|---|---|---|
| Solohead farm | Ireland | 150 | 52°51´N | 08°21´W | 10.6 | 1017 | 2004-2011 | (Necpálová et al., 2013) https://www.teagasc.ie/animals/dairy/research-farms/solohead/ |

**Table B2** Soil properties and management type of the grassland study sites

| Site | Grassland type | Management | Stocking rate | Total N fertilisation (kg N ha⁻¹ yr⁻¹) | Initial SOC in the topsoil (Mg C ha⁻¹ yr⁻¹) | Soil texture (%) Sand Silt Clay | Grass type |
|---|---|---|---|---|---|---|---|
| Laqueuille | Intensive semi-natural grassland | Grazing | ~1 | 210 | 114 | 27 53 20 | Grass clover mixture |
| Oensingen | Intensive | Cutting | - | 214 | 64.7 | 24 33 43 | Grass clover mixture |
| Easter Bush | Intensive | Grazing | 0.83 | ~ 229 | 93.26 | Clay loam (with | >99% rye grass (*Lolium Perenne*) |





| | | | | | | 20 to 26 % clay) | and < 0.5% white clover (*Trifolium repens*) |
|---|---|---|---|---|---|---|---|
| Solohead farm | Intensive, permanent grassland | Grazing | ~2 | ~183 | 137 | 25<br><br>33<br><br>42 | rye grass<br><br>and white clover<br><br>(20 to 25%) |

2.  *Input data for the model and main assumptions*

Average monthly temperature and precipitation for Laqueuille, Oensingen and Easter Bush sites were obtained from onsite

Meteorological Stations for the periods 2004-2012, 2004-2011 and 2004-2010 respectively. For Solohead dairy farm, climatic

data were provided by the Irish Meteorological Service referring to the nearest synoptic station with available climatic data for

the simulation period 2004-2011.  Monthly potential evapotranspiration was estimated using Thornthwaite equations

(Thornthwaite, 1948) in case of non-availability of data.

Plant carbon inputs in the different sites were estimated depending on the available data using the method described in the

section "Plant residues: Components and quality". For the Laqueuille site, average above-ground C residue was obtained from

available measured data and it represented 20% of above-ground C standing biomass (Table B3). We used the R:S ratio to

estimate below-ground biomass from average measured above-ground standing biomass. Below-ground C residues were

assumed to be 50% of the below-ground C biomass (Poeplau et al., 2016) (Table B3). For the Oensingen site, average above-

and below-ground C biomass were obtained from Ammann et al. (2009). We used the same assumption as Poeplau (2016) for

cutting grasslands, assuming that 30% of the above-ground biomass is not harvested, and that only 50 % of that fraction is

turned over annually and becomes available for soil organic matter formation (Schneider et al. 2006) (Table B3). To estimate

below-ground C residue, we used the same assumption as commented for Laqueuille site (Table B3). The same assumptions

were considered for the grazing Easter Bush site. From the average measured above-ground biomass we assumed only 20%

as residues as in the Laqueuille grazing site and the same hypothesis for the below-ground C residue as in the other previous

sites (Table B3). For Solohead dairy research farm, we referred to available measured data of above- and below-ground C





residues, using the same assumption for below-ground C residues as all the previous sites (Table B3). Finally, for the rhizodeposition as commented previously, we used an annual net rhizodeposition-to-root ratio of 0.5.

The proportions of plant C input added to the soil in each month were set as the pattern of inputs typical of European grasslands

suggested by Smith et al. (2005). Referring to plant residue quality we ascribed RPM and DPM pools related to NDF and ADF, respectively for each plant residue component (as described in the sub-section "Plant residues: components and quality").

The C animal excreta in Laqueuille grazing grassland were derived from Vertès et al. (2019) referring to the C intake grass-based rations, as the management is a continuous grazing from May to end of October without additional feed supply (Klumpp et al., 2011). Therefore, we estimated the C animal excreta as 32 % of the measured C intake using average values for the

simulation period 2004-2012. Annual C derived from cattle slurry in Oensingen site were estimated from Ammann et al. (2007) as an average of the provided years. Carbon input from grazing animal excreta was estimated the same as in Laqueuille site, while annual C input derived from organic fertilisation for Easter Bush was deduced from Jones et al. (2016) during the period 2004-2010 as 0.32 Mg C ha$^{-1}$yr$^{-1}$. The same method was used to estimate annual total N fertilisation and annual stocking rate of this site. For Solohead dairy research farm, C input derived from animal excreta were calculated the same as in

Laqueuille site and all other input data were estimated as average annual from the same study (Necpálová et al., 2013).

**Table B3** Plant C input of the different grassland intensive sites

| Plant C input in intensive grazing grassland Laqueuille site | |
|---|---|
| N input (Kg N ha$^{-1}$year$^{-1}$) | R:S ratio |
| 210 | 1,92 |
| Above-ground C (t C ha$^{-1}$) | Below-ground C (t C ha$^{-1}$) |
| 0,89 | 1,71 |
| Above-ground C residue (t C ha$^{-1}$) | Below-ground C residue (t C ha$^{-1}$) |





| 0,19 | 0,86 |
|---|---|
| Plant C input (above-and below-ground residues) | Net rhizodeposition (t C ha⁻¹) |
| 1.05 | 0.86 |
| **Plant C input in Oensingen cutting grassland site** | |
| Above-ground C (t C ha⁻¹) | Below-ground C (t C ha⁻¹) |
| 1.3 | 1,9 |
| Above-ground C residue (t C ha⁻¹) | Below-ground C residue (t C ha⁻¹) |
| 0,20 | 0,95 |
| Plant C input (above-and below-ground residues) | Net rhizodeposition (t C ha⁻¹) |
| 1,15 | 0.95 |
| **Plant C input in intensive grassland grazing site of Easter Bush** | |
| N input (Kg N ha⁻¹year⁻¹) | R:S ratio |
| 229 | 1,77 |
| Above-ground Biomass (t DM ha⁻¹) | Below-ground Biomass (t DM ha⁻¹) |
| 1.1 | 1.95 |
| Above-ground C (t C ha⁻¹) | Below-ground C (t C ha⁻¹) |
| 0,5 | 0.88 |
| Above-ground C residue (t C ha⁻¹) | Below-ground C residue (t C ha⁻¹) |
| 0,1 | 0,44 |
| Plant C input (above-and below-ground residues) | Net rhizodeposition (t C ha⁻¹) |



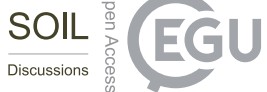

| 0.54 | 0.44 |
|---|---|
| **Plant C input of Solohead Research farm** | |
| Above-ground C residue (t C ha$^{-1}$) | Below-ground C residue (t C ha$^{-1}$) |
| 0,9 | 2.1 |
| Plant C input (above-and below-ground residues) | Net rhizodeposition (t C ha$^{-1}$) |
| 3 | 2.1 |

### C. Modified model performance

**Table C1** Model performance measurement indices

| Performance measure | Equation | Unit | Value range and purpose |
|---|---|---|---|
| BIAS, mean difference of simulations and observations (Smith and Smith, 2007) | $BIAS = \bar{P} - \bar{O}$ | Unit of the variable | negative to positive infinity: the closer the values are to 0, the better the model (negative values: underestimation; positive values: overestimation) |
| RMSE, Root Mean Square Error (Smith and Smith, 2007) | $RMSE = \frac{100}{\bar{O}} \times \sqrt{\frac{\sum_{i=1}^{n}(Oi-Pi)^2}{n}}$ | % | 0 to positive infinity: the closer the values are to 0, the better the model |
| EF, Model efficiency (Smith and Smith, 2007) | $EF = 1 - \frac{\sum_{i=1}^{n}(Pi-Oi)^2}{\sum_{i=1}^{n}(Oi-\bar{O})^2}$ | - | Negative infinity to 1 (optimum): the closer the values are to 1, the better the model |





P, predicted value; O, measured value; n, number of P/O pairs; i, each of P/O pairs; O, mean of measured values; P, mean of predicted values.

### D. Sensitivity analysis

**Table D1** NDF range of perennial ryegrass

| NDF min | NDF max | Reference |
|---------|---------|-----------|
| 48,9 | 52,2 | (Boudon and Peyraud, 2001) |
| 36,5 | 53,5 | (De Boever et al., 2013) |
| 40,7 | 49,4 | (Elgersma and Søegaard, 2016) |
| 45,8 | 58,5 | (Ergon et al., 2016) |
| 43 | 52,7 | (Frandsen, 1986) |
| 47,8 | 57,4 | (Küchenmeister et al., 2013) |
| 41,1 | 54 | (Lee et al., 2002) |
| 44 | 63.2 | (Ohlsson et al., 2007) |
| 39 | 53 | (Purcell et al., 2012) |
| 41,4 | 69,3 | (Salama et al., 2017) |
| 38.7 | 42.5 | (Van Vuuren et al., 2000) |
| 39.4 | 57.8 | (Armstrong et al., 1986) |
| 46.1 | 55.2 | (Østrem et al., 2014) |
| 32 | 47 | (Salama et al., 2012) |
| 49.8 | 57.4 | (Sun et al., 2010) |

**Table D2** Sensitivity index of varying resistant plant residues fraction from its minimum to maximum values in RothC_M for the different study sites

| Site | Output (min value) | Output (max value) | Sensitivity index |
|------|--------------------|--------------------|-------------------|
| Laqueuille | 118,6 | 120,4 | 1,5% |

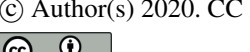



| | | | |
|---|---|---|---|
| Oensingen | 67,2 | 69 | 2,6% |
| Easter Bush | 87,6 | 88,4 | 0,8% |
| Solohead | 143,8 | 147,6 | 2,6% |

**Table D3** Sensitivity index of varying lignin content corresponding to animal excreta quality from its minimum to maximum values in RothC_M for the different study sites under current C input quantity (derived from animal excreta) and C input quantity (derived from animal excreta) scenario of 2.5 t C ha$^{-1}$ year$^{-1}$

| Site | Output (min value) | Output (max value) | Sensitivity index (%) |
|---|---|---|---|
| Laqueuille | 119.1 | 120.4 | 1.1 |
| Oensingen | 68.0 | 69.0 | 1.4 |
| Easter Bush | 87.3 | 88.7 | 1.6 |
| Solohead farm | 143.6 | 148.1 | 3.0 |
| Laqueuille (2.5 t C ha$^{-1}$) | 127.0 | 133.1 | 4.7 |
| Oensingen (2.5 t Cha$^{-1}$) | 74.2 | 79.5 | 6.6 |
| Easter Bush | 91.7 | 96.8 | 5.3 |

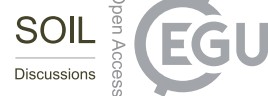

| (2.5 t C ha⁻¹) | | | |
|---|---|---|---|

**Table D4** Sensitivity index of varying the rate modifying factor for moisture from its minimum to maximum values in

RothC_M for the different study sites.

| Site | Output (max value) | Output (min value) | Sensitivity index |
|---|---|---|---|
| Laqueuille | 120 | 104,6 | 12,8% |
| Oensingen | 69,7 | 61,6 | 11,6% |
| Easter Bush | 89,6 | 85,3 | 4,8% |
| Solohead | 150,4 | 139,4 | 7,3% |






**Author contribution**

Asma Jebari

Conceptualization-Equal, Data curation-Lead, Formal analysis-Lead, Investigation-Lead, Methodology-Lead, Software-Equal, Writing-original draft-Lead

Jorge Álvaro-Fuentes

Conceptualization-Lead, Investigation-Lead, Supervision-Lead, Validation-Lead, Writing-review & editing-Lead

Guillermo Pardo

Methodology-Supporting, Software-Lead, Supervision-Supporting, Writing-review & editing-Equal


Maria Almagro

Formal analysis-Supporting, Methodology-Supporting, Resources-Supporting, Supervision-Supporting, Visualization-Equal, Writing-review & editing-Equal

Agustin Del Prado

Conceptualization-Lead, Funding acquisition-Lead, Investigation-Lead, Project administration-Lead, Supervision-Lead, Validation-Lead, Writing-review & editing-Lead

**Competing interests**

The authors declare that they have no conflict of interest.

**Acknowledgements**

We gratefully acknowledge the financial support of the Fundación Cándido de Iturriaga y Maria del Dañobeitia, Juan de la Cierva and the European Union's Horizon 2020 Research and Innovation Action (RIA) through the project "Innovation for sustainable sheep and Goat production in Europe (iSAGE)" undergrant agreement No 679302. BC3 is supported by the Basque Government through the BERC 2018-2021 program and by Spanish Ministry of Economy and Competitiveness MINECO

through BC3 María de Maeztu excellence accreditation MDM-2017-0714. Agustin del Prado is financed by the programme Ramon y Cajal from the Spanish Ministry of Economy, Industry and Competitiveness (RYC-2017-22143). María Almagro was supported by the Juan de la Cierva Program (grant IJCI-2015-23500). Finally, we thank Dr. Katja Klumpp and Dr. Stephanie Jones for providing useful information on field sites.





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
