# Peer review of "Simulating soil organic C dynamics in managed grasslands under humid temperate climatic conditions"

_SOIL, 2020_

## Referee Comment (RC1) · Anonymous Referee #1 · 23 Dec 2020

Recommendation: Reject

Comments 1. The design of the study is not convincing The authors proposed modifications for the Rothamsted Carbon (RothC) model for an improved prediction of soil organic carbon (SOC) dynamics in grassland soils. The proposed modifications are (i) change of the soil water function, (ii) use of the Van Soest fractionation for the estimation of pools (p. 4 to 5), (iii) separation of plant residue inputs into three components, and (iv) accounting for the animal treading effect. The authors present some interesting ideas for improvements. Unfortunately, there was no experimental design developed for such potential improvements of the RothC model. For instance, the was

no experiment devoted to improvements of the soil water function (e.g. using different water contents) or to improvements of pool estimations (e.g. using isotopes and fractionation). Overall, the importance of the different modifications cannot accurately be studied with the experimental design used.

2. The data basis of the study is too weak Unfortunately, the modifications were tested only on eight data points in total (initial and final SOC stocks for four field experiments, Fig. 2). This is not sufficient. Any overestimation of the SOC stock caused by one of the four modification may be balanced by an underestimation due to another modification.

3. The parameterization is not convincing Unfortunately, the authors use the IOM estimation equation by Falloon et al. (1998) which is based only on total SOC stocks. This IOM estimation adds additional uncertainty to the modelled SOC dynamics.

---

## Referee Comment (RC2) · Anonymous Referee #2 · 29 Dec 2020

The manuscript soil-2020-76 "Simulating soil organic C dynamics in managed grasslands under humid temperate climatic conditions" aimed to improve the prediction of SOC dynamics in managed grasslands under temperate climate conditions by uses of RothC model. To run Roth C under defined conditions, the SOC model was recalibrated to account for: (1) water content up to saturation conditions in the soil water function , (2) entry pools that account for particularity of exogenous organic matter (EOM) such as ruminant excreta), (3) annual variation in the carbon inputs derived from plant residues considering both above- and below-ground plant residue and rhizodeposits components as well as their quality, and (4) the livestock trampling (i.e., poaching damage).  the mode was than evaluated against four existing field experi-

ments in Europe. Analyses show a good model performance when implementing the four modifications. A higher sensitivity to soil moisture and plant residues modifications was observed compared to other modifications when grasslands were under intense grazing regime. Overall analyses suggest that RothC humid grassland modifications are applicable for farm and regional SOC dynamics from managed grassland-based systems.

The manuscript is well written, understandable and to my opinion definitely improves RothC for grazed grasslands (or at least the four tested). Given that RothC is originally a crop model, which has been improved for different crop residues and organic amendments, a "grazed grassland version" is most well, come. To be published, the present version would need some improvements to help the reader to get through the model modification and validations (in the moment in the supplementary material).

I thus recommend "revisions ".

General comments Tests of model-modifications (Table2) though I understand that modification where added on top of the other, I was wondering if they were tested individually and combined as some might go together e.g. (water saturation and poaching), (excreta and plant residues) see L359. I suggest to add a table on model performance on individually. (e.g. 2.3.4 and table 4) Along the some lines, I also wonder how the model (versions) were evaluated on SOC data, as most sites do not provide more than 2 to 3 soil sampling dates. The MM does not mention the tested data , see also comment just after.

The manuscript deals with the quality of plant residue and residues inputs by belowground biomass. However, the reader does not get any information on the tested sites !!!! They might all be the same. During model performance and sensitivity , these lack of basic information is misleading I thus strongly recommend to move site tables B1/B2/B3 to the main text, and to add main variables for the tested sites, so that reader can follow the model improvements/modification. I also suggest to add the basic columns to

differentiate the sites such as (tables B2/B3) i) temporary and permanent grasslands, ii) mowing and grazing and intensity iii) biomass production and biomass removal, iv) Root/Shoot, v) biomass quality (i.e. digestibility) .... These variables are used later on to evaluate.

This is also important to understand model sensitivity and sensitivity analyses. (eg L410ff,) as reader has only little idea on the field sites and grazing animals it is difficult to follow the mode performance. E. C inputs via animal dejections are result of stocking density and animal weight. Accordingly, there is difference between sheep and cattle. ... I suggest to add more information on sites and data inputs to MM section (i.e. tables B1 to B3 and texte L510 to L535 ).

It might also be interesting/useful to add tables of the sensitivity analyses to the main text. Eg. merger D2 to D4 for the different variables (leading to 10 column in total )

PLEASE find more specific comments to the texte in the here added supplement.zip file, including a .docx "revision mode" version of manuscript lines.

Please also note the supplement to this comment:
https://soil.copernicus.org/preprints/soil-2020-76/soil-2020-76-RC2-supplement.zip

---

## Author Comment (AC1) · 31 Dec 2020

Answer to comment 1

We agree with reviewer 1 that, ideally, the best way to modify and construct a model would be by using new large and high-quality datasets (e.g. long-term experiments, best-tech and sample numbers). However, since specific data (e.g. isotopes and fractionation that account for changes in vegetation growth) on grasslands under humid conditions seem to be missing, or at least not available, to our knowledge, we explored, as a hypothesis, the potential value of constructing small changes in the Roth-C model based on: i) changes already tested in other similar models (e.g. soil moisture function

by ECOSSE), ii) the scientific literature to account for particularity of exogenous or-
ganic matter (EOM) such as ruminant excreta and for the different components of plant
residues and distinguish its quality (iii) and available experiments under humid tem-
perate conditions to add a poaching effect sub-model. In the manuscript, we highlight
that the poaching modification is rather uncertain because of lack of long-term exper-
iments in the scientific literature (line 350, line 387 and line 430). The paper includes
a sensitivity analysis of the modified model to assess the robustness of the different
modifications (See "sensitivity analysis" section). In general, the model is more sen-
sitive to C input quantity than to quality and to soil moisture function, particularly at
saturation conditions.

Answer to comment 2

As previously mentioned, we agree that the more quality data used for a validation
exercise the better to prove the validity of a model within certain environmental and
management boundaries. Unfortunately, only four field studies have been found with
data availability. We tried to obtain data from at least three more sites (e.g., Dripsey in-
tensive grassland site and Carlow grassland site in Ireland, Haller research farm in the
USA. . .) but we could not get hold of these data. The validation of the model improve-
ments were made with the field experiments in humid temperate conditions included
in the FLUXNET program (http://www.fluxnet.ornl.gov/) (Baldocchi, 2008), which have
been presented in many studies (e.g., Ammann et al. 2009; Klumpp et al. 2011; Skiba
et al. 2013. . .) and their data are considered of high quality and have been used in
many studies before (e.g., Soussana et al. 2007; Ben Touhami et al. 2013; Sán-
dor et al. 2017. . .). We opted for grassland sites that are under temperate climatic
conditions (with precipitation > 1000 mm) and management regimes that are common
for intensive grassland-based livestock systems in this agroclimatic region of Europe:
Laqueuille intensive grazing grassland (France), Oensingen intensive cutting grass-
land (Switzerland), Easter Bush intensive grazing grassland (United Kingdom) and
Solohead dairy research farm (Ireland) because they presented similar climate and

management conditions (managed grasslands under humid temperate conditions). All these four experimental sites presented initial and end SOC stock values which were used in our model validation. For instance, Vuichard et al. (2007) improved PASIM model and tested it against measurements of 3 sites. Nemo et al. (2017) tested RothC initialisation referring to 4 sites. We understand that this validation exercise is useful but certainly, is not definite. This is recognised in the manuscript (validating against more sites would greatly improve the confidence of the model) (line 351, line 364, line 383, line 385, line 392, line 433).

Answer to comment 3

We agree that, ideally, the best way to get the IOM pool of RothC model would be from soil radiocarbon-SOC measurements. This is indeed something that the RothC developers indicate for example in Fallon et al. (2000). Since radiocarbon measurements are costly and rarely performed routinely, the same RothC developers indicated that IOM can be estimated, alternatively, from an empirically-derived relationship between IOM and total SOC (Falloon et al., 1998), which showed good fit (Falloon et al., 2000, 2006). The estimation of IOM with the equation proposed by Falloon et al (1998) has, in fact, been used in almost all the RothC modelling studies (e.g., Giongo et al., 2020; Francaviglia et al., 2012; Mondini et al., 2012. . .).

References

Ammann, C., Spirig, C., Leifeld, J. and Neftel, A.: Assessment of the nitrogen and carbon budget of two managed temperate grassland fields, Agric. Ecosyst. Environ., 133(3–4), 150–162, doi:10.1016/j.agee.2009.05.006, 2009.

Baldocchi, D.: 'Breathing' of the Terrestrial Biosphere: Lessons Learned from a Global Network of Carbon Dioxide Flux Measurement Systems, Aust. J. Agric. Res., 1–82, 2008.

Falloon, P., Smith, P., Coleman, K. and Marshall, S.: How important is inert organic

matter for predictive soil carbon modelling using the Rothamsted carbon model?, Soil Biol. Biochem., 32(3), 433–436, doi:10.1016/S0038-0717(99)00172-8, 2000.

Falloon, P., Smith, P., Bradley, R. I., Milne, R., Tomlinson, R., Viner, D., Livermore, M. and Brown, T.: RothCUK - A dynamic modelling system for estimating changes in soil C from mineral soils at 1-km resolution in the UK, Soil Use Manag., 22(3), 274–288, doi:10.1111/j.1475-2743.2006.00028.x, 2006.

Francaviglia, R., Coleman, K., Whitmore, A. P., Doro, L., Urracci, G., Rubino, M. and Ledda, L.: Changes in soil organic carbon and climate change - Application of the RothC model in agro-silvo-pastoral Mediterranean systems, Agric. Syst., 112, 48–54, doi:10.1016/j.agsy.2012.07.001, 2012.

Giongo, V., Coleman, K., da Silva Santana, M., Salviano, A. M., Olszveski, N., Silva, D. J., Cunha, T. J. F., Parente, A., Whitmore, A. P. and Richter, G. M.: Optimizing multifunctional agroecosystems in irrigated dryland agriculture to restore soil carbon – Experiments and modelling, Sci. Total Environ., 725(April), 138072, doi:10.1016/j.scitotenv.2020.138072, 2020.

Klumpp, K., Tallec, T., Guix, N. and Soussana, J. F.: Long-term impacts of agricultural practices and climatic variability on carbon storage in a permanent pasture, Glob. Chang. Biol., 17(12), 3534–3545, doi:10.1111/j.1365-2486.2011.02490.x, 2011.

Mondini, C., Coleman, K. and Whitmore, A. P.: Agriculture , Ecosystems and Environment Spatially explicit modelling of changes in soil organic C in agricultural soils in Italy , 2001 – 2100 : Potential for compost amendment, "Agriculture, Ecosyst. Environ., 153, 24–32, doi:10.1016/j.agee.2012.02.020, 2012.

Nemo, Klumpp, K., Coleman, K., Dondini, M., Goulding, K., Hastings, A., Jones, M. B., Leifeld, J., Osborne, B., Saunders, M., Scott, T., Teh, Y. A. and Smith, P.: Soil Organic Carbon (SOC) Equilibrium and Model Initialisation Methods: an Application to the Rothamsted Carbon (RothC) Model, Environ. Model. Assess., 22(3), 215–229,

doi:10.1007/s10666-016-9536-0, 2017.

Sándor, R., Barcza, Z., Acutis, M., Doro, L., Hidy, D., Köchy, M., Minet, J., Lellei-Kovács, E., Ma, S., Perego, A., Rolinski, S., Ruget, F., Sanna, M., Seddaiu, G., Wu, L. and Bellocchi, G.: Multi-model simulation of soil temperature, soil water content and biomass in Euro-Mediterranean grasslands: Uncertainties and ensemble performance, Eur. J. Agron., 88, 22–40, doi:10.1016/j.eja.2016.06.006, 2017.

Skiba, U., Jones, S. K., Drewer, J., Helfter, C., Anderson, M., Dinsmore, K., McKenzie, R., Nemitz, E. and Sutton, M. A.: Comparison of soil greenhouse gas fluxes from extensive and intensive grazing in a temperate maritime climate, Biogeosciences, 10, 1231–1241, doi:10.5194/bg-10-1231-2013, 2013.

Soussana, J. F., Allard, V., Pilegaard, K., Ambus, P., Amman, C., Campbell, C., Ceschia, E., Clifton-Brown, J., Czobel, S., Domingues, R., Flechard, C., Fuhrer, J., Hensen, A., Horvath, L., Jones, M., Kasper, G., Martin, C., Nagy, Z., Neftel, A., Raschi, A., Baronti, S., Rees, R. M., Skiba, U., Stefani, P., Manca, G., Sutton, M., Tuba, Z. and Valentini, R.: Full accounting of the greenhouse gas (CO2, N2O, CH4) budget of nine European grassland sites, Agric. Ecosyst. Environ., 121(1–2), 121–134, doi:10.1016/j.agee.2006.12.022, 2007.

Ben Touhami, H., Lardy, R., Barra, V. and Bellocchi, G.: Screening parameters in the Pasture Simulation model using the Morris method, Ecol. Modell., 266, 42–57, doi:10.1016/j.ecolmodel.2013.07.005, 2013.

Vuichard, N., Soussana, J. F., Ciais, P., Viovy, N., Ammann, C., Calanca, P., Clifton-Brown, J., Fuhrer, J., Jones, M. and Martin, C.: Estimating the greenhouse gas fluxes of European grasslands with a process-based model: 1. Model evaluation from in situ measurements, Global Biogeochem. Cycles, 21(1), 1–14, doi:10.1029/2005GB002611, 2007.

---

## Author Comment (AC2) · 29 Jan 2021

We would like to thank the reviewer for the consideration of our work and for requesting to individually respond to all referee comments. The authors would like to acknowledge the valuable points mentioned and the great feedback received from the reviewer's comments, which would help to improve the paper's readability and potential relevance for the scientific community.

Although we understand that this is not mandatory at this stage, we considered all suggested modifications and they were dealt with and partly or totally included to a new version of the manuscript.

**General comments**

Original comment

**Tests of model-modifications (Table2) though I understand that modification where added on top of the other, I was wondering if they were tested individually and combined as some might go together e.g. (water saturation and poaching), (excreta and plant residues) see L359. I suggest to add a table on model performance on individually. (e.g. 2.3.4 and table 4).**

➔ Response

In the original version of the manuscript, we did not test individually the different modifications. We incremented successively the different modifications in order to assess the progressive performance and improvement to the model. On reflection to reviewer's suggestion, we agree that an individual test could add extra information to our analysis of the model performance. In this new version, we have included a new table (Table 6 in the manuscript) which shows model performance individually. The new table is shown below and it illustrates the RMSE (i.e., Root mean square error) and BIAS (i.e., mean difference) of simulations and observations for each modification to the model and grassland intensive site and EF (i.e., model efficiency) and RMSE for each modification across sites.

**Table 6** Root mean square error (RMSE) and mean difference of simulations and observations (BIAS) for each modification to the model and grassland intensive site and model efficiency (EF) and RMSE across sites.

| Site | Performance | Soil moisture up to saturation | Ruminant excreta quality | Plant residue | Poaching |
|------|------------|-------------------|-----------------|--------------|----------|
| Laqueuille | BIAS | -7.27 | -18.45 | -16.88 | -18.91 |
| Oensingen | BIAS | -2.95 | -8.86 | -6.94 | - |
| Easter Bush | BIAS | -1.29 | -3.52 | -3.02 | -4.21 |
| Solohead | BIAS | -7.17 | -11.13 | -8.27 | -13.21 |
| All sites | RMSE | 5.51 | 10.75 | 9.19 | 12.19 |

| | EF | 0.95 | 0.80 | 0.86 | 0.46 |
|---|---|---|---|---|---|

Moreover, a discussion paragraph related to the added table was included to the manuscript (Lines 390-401).

*"Testing the model performance based on each of the individual modifications for the different sites allowed improving our understanding of its impact to the model (Table 6). Soil moisture up to saturation conditions in the soil water function of RothC showed the best performance compared with the other modifications (Table 6). The modification of RothC water function at saturation conditions fit to the temperate moist climatic conditions, since the different study sites showed saturation conditions most of the year. However, the poaching effect alone contributed to reduce SOC stocks and thus the model performance, since the poaching effect is related to water saturation conditions (Tuohy et al., 2014). The enhancement in the model performance showed by the quality of ruminant excreta depends on its quantity. Indeed, the BIAS reduction with ruminant excreta quality modification compared with the default version (Table 5 and Table 6) was more important in the grassland sites with major ruminant excreta application (e.g., Solohead research farm). However, the plant residue modification showed a higher improvement compared with the ruminant excreta quality as it implies an increase in C inputs with the inclusion of the rhizodeposition component. The latter was recommended to be added to the different plant residue components in SOC models (Rumpel and Kögel-knabner 2011) especially RothC (Balesdent et al. 2011)."*

Furthermore, as suggested by the Reviewer, we tested the model based on the combined effect of (i) water saturation and poaching and (ii) excreta and plant residues. In the new version, we have introduced a new table (Table 7 in the manuscript) which shows the performance of the combined modifications (i.e., soil moisture up to saturation and poaching; ruminant excreta and plant residues) to the model, as illustrated in the table below.

**Table 7** Root mean square error (RMSE) and mean difference of simulations and observations (BIAS) for the combined modifications (soil moisture up to saturation and poaching; ruminant excreta and plant residues) to the model and grassland intensive site and model efficiency (EF) and RMSE across sites.

| Site | Performance test | Soil moisture saturation and Poaching effect | Ruminant excreta and plant residues |
|---|---|---|---|
| Laqueuille | BIAS | -7.79 | -16.56 |
| Oensingen | BIAS | -2.95 | -6.58 |
| Easter Bush | BIAS | -1.44 | -2.43 |
| Solohead | BIAS | -7.96 | -6.88 |
| All sites | RMSE | 5.96 | 8.66 |

| | | |
|---|---|---|
| EF | 0.94 | 0.87 |

A paragraph discussing the importance of the different modifications was added to the new version (Lines 405-414).

*"However, testing the model based on the combined effect of soil moisture up to saturation and poaching effect showed a great performance compared with excreta and plant residues with a RMSE of 5.96 compared with 8.66 (Table 7). The modifications of soil moisture up to saturation and poaching effect reduced the BIAS compared with animal excreta and plant residue modifications for the different study sites, except for the Solohead research farm. This could be explained by the fact that the latter received higher C inputs derived from animal excreta and plant residues and lower water saturation conditions compared with the other sites (Table 2). Therefore, the model modification with the greatest positive impact was soil moisture up to saturation (Table 6 and Table 7). However, the impact of plant residues and ruminant excreta modifications depend on the C input quantity (Table 6 and Table 7). The poaching effect could not be considered without taking into account the soil moisture saturation modification, as it showed a lower performance than the default model version (Table 5 and Table 6)."*

Original comment

**Along the some lines, I also wonder how the model (versions) were evaluated on SOC data, as most sites do not provide more than 2 to 3 soil sampling dates. The MM does not mention the tested data, see also comment just after.**

➢ Response

We agree with the reviewer that, having more soil-sampling dates would have been great to increase the robustness of our validation exercise. We opted for four grassland sites: Laqueuille intensive grazing grassland, Oensingen intensive cutting grassland, Easter Bush intensive grazing grassland and Solohead dairy research farm because they presented the conditions of intensive management under temperate climate, with precipitation > 1000 mm. With the aim to get a balanced comparison among the sites used in our model validation, we opted for initial and end average SOC stock values, since they were commonly available for the different study sites. But, according to Reviewer's comment, in the new version of the manuscript, we present all available SOC sampling data (two to four annual soil measured data) for each experiment. Indeed, we have contacted the authors of the different studies used and we did not find more annual SOC data for each of the study sites. Particularly, we added SOC measured data for Laqueuille intensive grassland site (total of three SOC measured dates) and Solohead research farm (total of four SOC measured dates) (See Table 5 below and Fig. 2).

*"In general, the highest predicted SOC stocks values and the closest to the measured values at the end of the simulation period resulted after RothC_3 and RothC_4 simulations (Fig. 2). For Laqueuille grassland intensive site, RothC_3 and RothC_4 were able to match the general trend of SOC increase (between 2004 and 2012) and the SOC stocks at the end of the simulation period, but not the change of SOC stocks corresponding to the year 2008 (Fig. 2). However, SOC*

*simulation for Solohead research farm, using RothC_3 and RothC_4 modified versions were within the range of measured data of SOC stocks (Fig. 2)."*

In general, adding all these extra points, the model performance was not significantly affected (Table 5).

**Table 5** Root mean square error (RMSE) and mean difference of simulations and observations (BIAS) for each model version and grassland intensive site and model efficiency (EF) and RMSE across sites.

| Site | Performance test | RothC_0 | RothC_1 | RothC_2 | RothC_3 | RothC_4 |
|------|------------------|---------|---------|---------|---------|---------|
| Laqueuille | BIAS | -18.77 | -7.27 | -6.95 | -4.26 | -4.55 |
| Oensingen | BIAS | -9.22 | -2.95 | -2.61 | 0.32 | 0.32 |
| Easter Bush | BIAS | -4.10 | -1.29 | -0.74 | 0.51 | 0.35 |
| Solohead | BIAS | -12.52 | -7.17 | -5.78 | -1.02 | -2.00 |
| All sites | RMSE | 11.36 | 5.51 | 4.86 | 2.77 | 3.01 |
| | EF | 0.78 | 0.95 | 0.96 | 0.99 | 0.98 |

Furthermore, we highlighted in the new manuscript version that validating against more sites would enhance the confidence to the model modifications (e.g., Lines 421 and 494).

*"Although RothC_3 and RothC_4 simulations performed well in simulating SOC changes for the selected sites, there were limitations related to the uncertainty of, both, model inputs and modifications, and to the limitation of the data used for validation."*

*"It must be kept in mind that although there was good agreement between results from modified model and measured data from different studies, validating against more sites would greatly improve our confidence."*

In the new manuscript, we also added information about the observed SOC stocks that we used in the simulations (Table 2).

Original comment

**The manuscript deals with the quality of plant residue and residues inputs by belowground biomass. However, the reader does not get any information on the tested sites !!!! They might all be the same. During model performance and sensitivity, these lack of basic information is misleading I thus strongly recommend to move site tables B1/ B2/B3 to the main text, and to**

add main variables for the tested sites, so that reader can follow the model improvements/modification. I also suggest to add the basic columns to differentiate the sites such as (tables B2/B3) i) temporary and permanent grasslands, ii) mowing and grazing and intensity iii) biomass production and biomass removal, iv) Root/Shoot, v) biomass quality (i.e. digestibility) …. These variable are used later on to evaluate.

This is also important to understand model sensitivity and sensitivity analyses. (eg L410ff,) as reader has only little idea on the field sites and grazing animals it is difficult to follow the mode performance. E. C inputs via animal dejections are result of stocking density and animal weight. Accordingly, there is difference between sheep and cattle…. I suggest to add more information on sites and data inputs to MM section (i.e. tables B1 to B3 and text L510 to L535).

➔ Response

In the new version of the manuscript, we included a new table (Table 2) with detailed information on the different study sites, as suggested by the reviewer. We also merged the tables dealing with the characterization of the different intensive temperate grasslands and moved the table to the "Materials and Methods" section as suggested in the specific comments (Table 2 in the new manuscript). Apart from plant biomass, plant residues and its different components (above- and below-ground residues and rhizodeposits), we included NDF variability for the different plant residue components as it refers to plant residue quality (i.e., RPM). In addition, we added information about the management type (temporary and permanent grasslands, mowing and grazing frequency, fertilization type and frequency…), as suggested. Available data on soil characterization (texture, drainage…) was also introduced, as well as the tested SOC data with the corresponding depth. We included C input data derived from ruminant excreta. The quality of this latter is illustrated in Table 1 of the new manuscript as we specified the excreta for ruminants in general to differentiate them from other animals like pigs…and we did not specify the excreta between the different ruminants.

All information on geographic and climatic characteristics, soil properties, input data and management of the different sites, suggested by the reviewer, is illustrated in the following table.

**Table 2** Location, climate, soil properties, management type and input data to the model of the grassland study sites (available through the European Fluxes Database Cluster: http://www.europe-fluxdata.eu (except Solohead farm)

| Site name and references | Laqueuille (Klumpp et al., 2011) | Oensingen (Ammann et al., 2009) | Easter Bush (Skiba et al., 2013; Jones et al., 2016) | Solohead farm (Necpálová et al., 2014) |
|---|---|---|---|---|
| Country | France | Switzerland | United Kingdom | Ireland |
| Altitude (m) | 1040 | 450 | 190 | 150 |
| Latitude | 45° 38´N | 47° 17´N | 55° 52´N | 52°51´N |
| Longitude | 02° 44´E | 07° 44´E | 03° 02´W | 08°21´W |
| Mean air temperature ($^O$C) | 7 | 9 | 9 | 10.6 |

| | | | | |
|---|---|---|---|---|
| Mean annual precipitation (mm) | 1012 | 1263 | 1031 | 1017 |
| Simulation period | 2004-2012 | 2004-2011 | 2004-2011 | 2004-2011 |
| Grassland type | Intensive semi-natural permanent grassland | Intensive permanent grassland | Intensive permanent grassland | Intensive permanent grassland |
| Management (Mowing/Grazing frequency) | -Grazing by heifers (May to October) | Grass mowing (4 times a year) No grazing | - Grazing all year round by cattle and sheep | Grazing by dairy cows February to November Mowing |
| Annual production (t ha$^{-1}$ yr$^{-1}$) | 7 | 7.5 | 5.6 | 13.5-14.7 |
| Stocking rate (LSU ha$^{-1}$ yr$^{-1}$) | ~1 | - | 0.83 | ~2 |
| Total N fertilisation | Mineral fertilisation in three splits: 210 kg N ha$^{-1}$ yr$^{-1}$ | Solid ammonium nitrate or liquid cattle manure at the beginning of each growing cycle (after the previous cut): 214 kg N ha$^{-1}$ yr$^{-1}$ | Ammonium nitrate fertiliser was applied to the field 3-4 times per year, usually between March and July (~ 229 kg N ha$^{-1}$ yr$^{-1)}$ | N fertiliser ~183 kg N ha$^{-1}$ yr$^{-1}$ applied from February to September |
| SOC in the topsoil (Mg C ha$^{-1}$ yr$^{-1}$) | 114 ± 1.48 (20 cm depth) in 2004 125.8 (20 cm depth) in 2008 121±2.35 (20 cm depth) in 2012 | 64.7 (20 cm depth) in 2004 68.3±1.6 (20 cm depth) in 2011 | 93.26 (30 cm depth) in 2004 87.87 (30 cm depth) in 2011 | 137±6.5 (30 cm depth) in 2004 142.8±7.14 (30 cm deth) in 2008 148.8±7.16 (30 cm deth) in 2009 149.2±9.7 (30 cm depth) in 2011 |

*(continue next page)*

| Site name and references | Laqueuille (Klumpp et al., 2011) | Oensingen (Ammann et al., 2009) | Easter Bush (Skiba et al., 2013; Jones et al., 2016) | Solohead farm (Necpálová et al., 2014) |
|---|---|---|---|---|
| Soil properties | The soil is an Andosol (20% clay, 53% silt and 27% sand) with 11% carbon and 18% organic matter. | The soil is classified as Eutri-Stagnic Cambisol (FAO, ISRIC and ISSS, 1998) developed on clayey alluvial deposits. Clay contents between 42% and 44% induce a total pore volume of 55% and a fine pore volume of 32% (permanent wilting point) | The soil type is an imperfectly drained Macmerry soil series,Rowanhill soil association (Eutric Cambisol) with a pH of 5.1 (in H$_2$O) and a clay fraction of 20-26%. | The predominant soils are poorly drained gleys (90%) and grey-brown podzolics (10%) with a clay loam texture and low permeability (28% clay, 35%silt) |
| Grass type | Grass clover mixtureThe dominant grass are *Dactylis glomerata, Trisetum flavescens, Poa pratensis and Agrostis capillaris* | Grass clover mixture | >99% rye grass (*Lolium Perenne*) and < 0.5% white clover (*Trifolium repens*) | rye grass and white clover (20 to 25%) |
| R:S ratio | 1.92 | 1.46 | 1.77 | |
| Above-ground C (t C ha$^{-1}$) | 0.89 | 1.3 | 0.5 | |
| Below-ground C (t C ha$^{-1}$) | 1.71 | 1.9 | 0.88 | |
| Plant residue components (t C ha$^{-1}$) | C$_a$ = 0.19; C$_b$=0.86; C$_r$=0.86 | C$_a$ = 0.2; C$_b$=0.95; C$_r$=0.95 | C$_a$ = 0.1; C$_b$=0.44; C$_r$=0.44 | C$_a$ = 0.9; C$_b$=2.1; C$_r$=2.1 |
| Biomass quality (NDF range) | NDF$_a$ ranges from 0.55 to 0.67 NDF$_b$ ranges from 0.63 to 0.75 NDF$_r$=0 | NDF$_a$ ranges from 0.56 to 0.68 NDF$_b$ ranges from 0.64 to 0.76 NDF$_r$=0 | NDF$_a$ ranges from 0.55 to 0.69 NDF$_b$ ranges from 0.63 to 0. 77 NDF$_r$=0 | NDF$_a$ ranges from 0.51 to 0.64 NDF$_b$ ranges from 0.59 to 0.72 NDF$_r$=0 |

| | | | | |
|---|---|---|---|---|
| C input derived from ruminant excreta (t C ha$^{-1}$ yr$^{-1}$) | 0.54 | 0.47 | 0.75 | 2.3 |

NDF$_a$, Neutral Detergent Fiber corresponding to resistant above-ground plant material; NDF$_b$, Neutral Detergent Fiber corresponding to resistant below-ground plant material; NDF$_r$, Neutral Detergent Fiber corresponding to rhizodeposits

C$_a$, above-ground plant C input; C$_b$, below-ground plant C input; C$_r$, plant C input corresponding to rhizodeposition.

Original comment

**It might also be interesting/useful to add tables of the sensitivity analyses to the main text. Eg. merger D2 to D4 for the different variables (leading to 10 column in total)**

➔ Response

We merged the tables of the sensitivity analysis and moved the modified table to the main text as suggested (Table 8 of the new manuscript). The modified table, dealing with the sensitivity of the modified model to the main modifications, is illustrated below.

**Table 8** Sensitivity index of varying resistant plant residues fraction, lignin content corresponding to animal excreta quality and the rate modifying factor for moisture from its minimum to maximum values in RothC_4 for the different study sites.

| | Plant residues quality (Resistant fraction) | | | Animal excreta quality (Lignin content) | | | Rate modifying factor for soil moisture | | |
|---|---|---|---|---|---|---|---|---|---|
| Site | Output (min) | Output (max) | Sensitivity index | Output (min) | Output (max) | Sensitivity index | Output (min) | Output (max) | Sensitivity index |
| Laqueuille | 118.6 | 120.4 | 1.5% | 119.1 | 120.4 | 1.1% | 104.6 | 120 | 12.8% |
| Oensingen | 67.2 | 69 | 2.6% | 68 | 69 | 1.4% | 61.6 | 69.7 | 11.6% |
| Easter Bush | 87.6 | 88.4 | 0.8% | 87.3 | 88.7 | 1.6% | 85.3 | 89.6 | 4.8% |
| Solohead | 143.8 | 147.6 | 2.6% | 143.6 | 148.1 | 3% | 139.4 | 150.4 | 7.3% |

The changes to the text suggested by the reviewer within the 'specific comments' section are very much valuable and will improve the model readability and relevance. All suggested modifications were included to a new version of the manuscript although it is not mandatory at this stage.

**Specific comments**

| Original Comment | Response | Outline of the change in the new manuscript |
|---|---|---|
| **L13 suggest to add some details here e.g. ….. while managed grasslands have received much less attention. Managed grassland do have particularities with respect to grazing animals leading to soil compaction, changes in vegetation growth and quality and animal dejections. In this regard, we aimed to improve the prediction of SOC dynamics in managed grasslands under temperate climate conditions.** | Details added as suggested | See Line 11 |
| **L14 RothC, originally developed to model the turnover of SOC in arable topsoils,**  | We got rid of the sentence as suggested | See Line 15 |
| **L19 the livestock trampling effect (i.e., poaching damage) as a common problem in humid areas with higher annual precipitation.** | We used "trampling" instead of "treading" as suggested | See Line 20 |
| **L20ff**  Here **we describe the basis of these modifications according to a simple sensitivity analysis and validate** model **predictions against data from** four  **field experiments**  **in Europe. Model performance showed that modified RothC**  **captures well the different modifications. However, the model** was  **more sensitive to soil moisture and plant residues…** | We introduced the suggested modifications | See Lines 21-25 |
| **L29 please add other than Soussana et al 2004.**  **Therefore, …** | We add two more references (Conant et al., 2017; Eze et al., 2018a). We got rid of the sentence "*Moreover, … (Mottet et al., 2017).*" | See Line 29 |
| **L48 to 59 move to L29** | We moved the paragraphs as suggested | See Lines 29-41 |

| | | |
|---|---|---|
| **L39….** to predict long-term responses of grasslands to  climate change and management (FAO, 2018). However, models vary in complexity depending on their fundamental objectives (Taghizadeh-toosi and Olesen, 2016)they have been developed for. | Sentences modified as suggested | See Lines 50-52 |
| **L44ff** Amongst these models, the RothC model, originally developed for arable soils, is one of the models that has been most widely validated and effectively used for different agricultural systems at different spatial scales (e.g. Poeplau and Don, 2013; Senapati et al., 2013; Smith et al., 2014).  | Sentence modified as suggested | See Line 55 |
| **L54** Furthermore, grazing and wheeling by vehicles can cause damage  soil and vegetation structure by trampling and poaching,  both affecting  plant production, and the potential amount of C inputs causing soil C loss. | Sentence modified as suggested | See Line 36 |
| **L60** Studies using RothC for grassland ecosystems require  specific initialization | We changed "implied" by "require" as suggested | See Line 58 |
| **L63ff** RothC indirectly simulates grazing activity by altering the amount  total plant C inputs, where plant residues do not differentiate between above- and below-ground C inputs (Nemo et al., 2017). As for animals C inputs, RothC offers default quality values for C inputs from grazing animals or manure applications, but it does not consider the soil compaction and other  soil physical conditions related to grazing (Smith et al., 2014).  Farina et al.(2013)have  reduced decomposition rate in soil to improve model performance under dryland conditions. However, for water-logged conditions RothC does not account for  humid saturated conditions which imply oxygen limitation and thus a decline in decomposition rate (Moyano et al., 2013). To adapte RothC of humid grazed grassland conditions  | All modifications were introduceed to the paragraph | See Lines 61-71 |

| | | |
|---|---|---|
|  we studied which of the aforementioned factors (i) could be easily implemented in RothC, (ii) does  affect SOC changes and (iii)  allows to improve RothC predictions of SOC changes. To evaluate model performance related to  modifications , model outputs were assessed against  published experiments  by using sensitivity analysis and a stepwise approach  | | |
| **L86ff**  **four modifications were proposed and tested in this study: (i)** **extensions** **of soil water content function**  **up to saturation; (ii)** **enlargement of carbon input** **pools to account for the diversity of applied exogenous organic matter (EOM) from ruminant excreta; (iii) affinition of plant residue components and quality variability; and (iv)** **the trampling****/poaching effect of grazing animals.** | Sentence modified as suggested | See Line 84 |
| **L90 RothC**  **contains a minimum rate modifying factor**  **when soil moisture is at**  **minimum moisture capacity (i.e., at the extreme of water limitation). However, no correction is applied under water saturation and when soil is oxygen limitated** | Sentence modified as suggested. The rate modifying factor when soil moisture is at the extreme of water limitation in the RothC model is of 0.2 | See Line 88 |
| **L94ff ….at saturation conditions, as suggested by Smith et al. (2010) in the ECOSSE model. The conversion from soil water content to soil moisture deficit (SMDi, mm) used in RothC is…**
 $SMD_i = (WC_i - W) \times 10 \times dept\,h$
 **(1)**
 **Where WC fc is the soil water content at field capacity, WC i is the soil water content above field capacity. Soil water contents at saturation and field capacity conditions are in turned estimated by considering soil properties related to soil texture as described by**
 **(Raes et al., 2017).** | Sentences modified as suggested | See Line 92 |
| **L108 …above-mentioned studies have summed up all the different animal excreta into one category and did not distinguish excretions from different animal types (e.g., ruminants, pigs…). In order to capture the specific characteristics of ruminant excreta, we developed a methodology based on Pardo et al. (2017) as illustrated in Fig. A1. In this study Pardo et al (2017) proposed a partition of the C inputs from excreta into RothC pools based the relationship between lignin content (Van Soest fractions) and anaerobic** | Changes were included as suggested | See Lines 104-110 |

| | | |
|---|---|---|
| biodegradability, estimated as follows (Eq. (2)): $B = 0.905 \times exp(-0.055 \times lig(\%))$ | | |
| L125 The Van Soest fractions were derived from literature review for every animal excreta type of ruminants. However, a large variability in animal excreta (Fig. A2, Fig. A3) was observed depending a number of factors, an in particular on diet (e.g., high concentrate diet implies lower lignin content in the ruminant´s excreta). In order to fit ruminant excreta quality to the RothC entry pools, we applied an average value for all fractions (Table 1). | Changes were included as suggested | See Lines 121-124 |
| L134ff ground residues as a surrogate for total plant C inputs and do account less for root inputs (Nemo et al., 2017). Here we separated the plant residue C inputs into three components (i.e., above-ground residues, below-ground residues and rhizodeposits). The structure of C input derived from plant residues in RothC modified model is as illustrated in Fig. A4. To partition biomass into aboveground and below-ground biomass , we used the root to shoot (R:S) (when its value is not available). We assumed N fertilisation as the main driver for R:S ratio in grasslands as many studies have proved the strong dependence of the latter on N inputs (Poeplau, 2016 and Sainju et al., 2017). We referred therefore to Poeplau (2016) equation (Eq. (6)) for RothC C input parameterisation under temperate grasslands in order to consider the fertilisation effect on the R:S ratio: $R{:}S = 4.7375\, e^{-0.0043 \cdot Ninput}$ | Changes were included as suggested | See Lines 130-137 |
| L150 Plant residue quality (biochemical composition), as one of the main drivers of decomposition, is generally included (add REF e.g. Kazakou et al 2006, Fortunel et al 2009) | We added the suggested references (Kazakou et al 2006, Fortunel et al 2009) | See Line 145 |
| L157 maturity stage, climate variables and nitrogen fertilisation). Add references! | We added the references (Ball et al., 2001; Buxton, 1996) | See Line 152 |
| L163…. NDS measured data, there are existing empirical existing equations that can help to have an estimation of these parameters. For our study we used an existing the equation from Salcedo (2015)… | We modified the sentences as suggested | See Lines 157-158 |
| L168 … Where CP is crude protein and is expressed as a percentage of dry matter (CP is variable and depends on the stage of plant growth. It was obtained according to grassland plant species and their growth stage); $T^a$ mean is the monthly air temperature in °C; Water | We included the suggested modifications | See Line 162 |

| | | |
|---|---|---|
| **reserves refer to the difference between monthly precipitation and potential evapotranspiration.** | | |
| **L178 2.2.4 Animal treading effect: Poaching Trampling?** | The sub-title was modified as suggested | See Line 173 |
| **L181 .. and its impact on plant C inputs depending on soil moisture, soil compaction and degradation under grazing conditions (i.e., stocking rate) (Fig.1). Soil** | Modified as suggested | See Line 176 |
| **L183…. According to Piwowarczyk et al. (2011) and Herbin et al. (2011), we used SMD as a proxy for soil moisture to predict when soil water conditions are likely to lead to hoof damage.** | We included the suggested modifications | See Line 178 |
| **L191…As the poaching effect in temperate grazing systems seems to cause only short-term pasture reduction NOT CLEAR reduction in what??** | As the poaching effect in temperate grazing systems seems to cause only short-term reduction in pasture plant production. | See Line 186 |
| **L197…. In order to validate the proposed modifications, we used data from four studies European grasslands having temperate conditions and being characterized by precipitations > 100mm (??) during growth periode, and grass and clover mixture. I think non of the grassland is defined humid.** | We introduced the modification to the sentence. The used study sites presented temperate conditions with annual precipitations higher than 1000 mm
We think that temperate moist climatic conditions might describe the climate type of the different experiments used in our study.
We changes then "humid" term by "moist" through the manuscript | See Line 194 |
| **Figure 2 colors are not easy to see and separate I suggest to use different** | We used different colors as suggested.
See attached document | Fig 2 in the new manuscript |
| **L345… A possible explanation to this improvement in the SOC predictions is that the soil in Easter Bush site is poorly drained …. THIS needs to be mentioned in the MM section. From table B1 the reader does not know the particularies of the sites.** | We mentioned in the "Materials and Methods" section the available information of soil characteristics for each site. Table 2 in this document | See Table 2 |
| **L356… In RothC_4, we considered different variables (i.e., soil texture, precipitation, grass type, grazing intensity, study duration and sampling depth)…. NOT detailed in table B1 the** | We detailed all the available information about the different study sites in Table 2 in the "Materials and | See Table 2 |

| | | |
|---|---|---|
| reader does not know the particularies of the sites. !!!! | Methods" section, as suggested.
Table 2 in this document | |
| L 359 … In contrast, the effect of animal excreta quality and poaching on SOC simulation by RothC was low. This is not surprising as the Oensingen site is mown ! Also authors did not tested combined effects which might go together e.g. (water saturation and poaching), (excreta and plant residues). | We tested the effect of each of the individual modification as well as the combined effect of the related modifications as suggested (See Table 6 and 7 and their related paragraphs in this document). | See Table 6 and Table 7
Lines 390 – 401
Lines 405 – 414 |
| L418 …. Sensitivity index regarding soil moisture modification was higher compared with the other modifications reaching, for example 12.8% in the Laqueuille site (Table D4). …. *I think this also linked to soil texture and soil type (such as andosol)* | We stressed out that "The variation in the sensitivity index, related to soil moisture modification, among the different study sites depend on their soil properties." | See Line 476 |
| L 424….To our knowledge, this study is the first attempt to integrate grassland growth (i.e. growth stages, R/S) and management effect such as grazing and accompanied conditions under  into the RothC model to simulate SOC changes. The proposed modifications to the model considered the incorporation of | We included the modifications, as suggested | See Lines 485-488 |
| L436ff ….The modifications presented here to the RothC model may improve assessments of SOC changes in managed grasslands under temperate climatic conditions not only at a plot level but also at regional level. As such RothC-grassland version might be a useful tool for stakeholders and policy makers in order to improve the quantification of SOC sequestration and develop effective strategies to reduce the impact of grassland-based livestock systems | Modifications were introduced, as suggested | See Lines 497-500 |
| L 490 The site was continuously grazed by heifers (1.1 SR/ha/yr) from May to October without feed supply, …….and fertilized ???? | We added the information about fertilization of the Laqueuille grassland intensive site.
… "*and fertilized with 210 kg N ha$^{-1}$year$^{-1}$ (ammonium nitrate) in three splits*" | See Line 544 |
| L493 …. fertilised beginning of each growing cycle …. HOW much and what | We added the information about fertilization of the | See Line 547 |

| | | |
|---|---|---|
| | Oensingen cutting grassland site

*.... "fertilised with 214 kg N ha$^{-1}$year$^{-1}$ (as solid ammonium nitrate or liquid cattle manure) at the beginning of each growing cycle."* | |
| **Merge tables B1 and B2 and B3 and move to MM add the text L510 to L535 to MM**
**Table B2 unit of stocking rates** | Tables were merged and moved to the "Materials and Methods" section.
We moved the paragraph to the "Materials and Methods" section.
We added the unit of stocking rate as LSU ha$^{-1}$ yr$^{-1}$

(Table 2) | See Table 2 and Lines 200-225 |

[Figure]

[Figure]

[Figure]

[Figure]

**Figure 2** Measured and simulated SOC stocks (Mg C ha$^{-1}$) using the default RothC model (RothC_0) and the modified RothC versions (RothC_1; RothC_2; RothC_3; and RothC_4) for the different validation sites: (a) Laqueuille intensive grazing grassland; (b) Oensingen intensive cutting grassland; (c) Easter Bush intensive grazing grassland; and (d) Solohead dairy research farm.